# DISCRETE VARIATIONAL AUTOENCODING VIA POLICY SEARCH

**Michael Drolet**[1,*] **Firas Al-Hafez**[1], **Aditya Bhatt**[1,2], **Jan Peters**[1,2,3,4] **& Oleg Arenz**[1]
[1] Technical University of Darmstadt   [2] German Research Center for AI (DFKI)
[3] Hessian.AI   [4] Robotics Institute Germany (RIG)
Darmstadt, Hesse, Germany

## ABSTRACT

Discrete latent bottlenecks in variational autoencoders (VAEs) offer high bit ef-
ficiency and can be modeled with autoregressive discrete distributions, enabling
parameter-efficient multimodal search with transformers. However, discrete ran-
dom variables do not allow for exact differentiable parameterization; therefore,
discrete VAEs typically rely on approximations, such as Gumbel-Softmax repa-
rameterization or straight-through gradient estimates, or employ high-variance
gradient-free methods such as REINFORCE that have had limited success on
high-dimensional tasks such as image reconstruction. Inspired by popular tech-
niques in policy search, we propose a training framework for discrete VAEs that
leverages the natural gradient of a non-parametric encoder to update the paramet-
ric encoder without requiring reparameterization. Our method, combined with
automatic step size adaptation and a transformer-based encoder, scales to chal-
lenging datasets such as ImageNet and outperforms both approximate reparame-
terization methods and quantization-based discrete autoencoders in reconstructing
high-dimensional data from compact latent spaces.

## 1 INTRODUCTION

Discrete representations play a critical role in numerous fields, including telecommunications, biol-
ogy, and robotics. While real-world data is often continuous, introducing a discrete latent bottleneck
can offer unique advantages, such as increased bit-efficiency, multimodal modeling, and the ability
to leverage tools from combinatorial optimization. These properties make discrete representations
particularly useful for tasks that benefit from structured or compact latent spaces. However, encod-
ing continuous information into discrete representations and decoding it back remains a challenging
endeavor. In particular, gradient-based optimization requires a differentiable transformation to map
continuous inputs to discrete latents, but this is impeded by the non-differentiability of operations
like rounding or taking the argmax, which are essential for obtaining hard assignments. Because of
this difficulty, we are often left with approximate techniques, such as Gumbel Softmax reparameter-
ization and straight-through esimation Jang et al. (2016).

However, approximate reparameterization with Gumbel-Softmax Jang et al. (2016) is sensitive to
its temperature hyperparameter, forcing a trade-off between increased gradient variance at low tem-
peratures and large approximation error at high temperatures. Furthermore, for larger bottlenecks,
backpropagating the soft-assignments through the autoregressive sampling process has high memory
footprint and can suffer from vanishing gradients. Variance-reduction schemes such as the Gumbel-
Rao Monte Carlo estimator (GR-MCK) Paulus et al. (2020) help but do not fully resolve the scaling
problem. On the other hand, vector-quantization methods, such as the Vector-Quantized Variational
Autoencoder (VQ-VAE) Van Den Oord et al. (2017) and Finite Scalar Quantization (FSQ) Mentzer
et al. (2023) avoid the problems of discrete reparameterization, but still rely on approximations
due to the use of straight-through estimation. Furthermore, their latent distribution is intractable,
which prevents them from maximizing the latent entropy in the Evidence Lower Bound (ELBO)
and demands special training objectives. Score-function-based gradient estimators, such as RE-
BAR (Tucker et al., 2017) or Muprop (Gu et al., 2015), are a promising alternative as they can
obtain unbiased gradients to optimize the ELBO in the discrete setting. However, due to their high
gradient variance, the success of these methods has so far been limited.

---

[*]Michael Drolet (`michael.drolet@tu-darmstadt.de`) is the corresponding author.
Code and supplementary material are available at `drolet.io/daps`.

To overcome these limitations, we propose Discrete Autoencoding via Policy Search (DAPS), a training framework that optimizes the ELBO for discrete, autoregressive encoders without any discrete reparameterization or straight-through tricks. DAPS casts encoder learning as a KL-regularized policy-search problem: we form a closed-form, nonparametric target distribution $q^*$ and update a parametric encoder $q_\theta$ by weighted maximum likelihood, avoiding backpropagation through the sampling path. A single scalar trust-region parameter $\eta$ is adapted automatically using an effective sample size (ESS) objective, yielding stable step sizes across tasks and loss scales. We additionally include a robotics experiment to demonstrate that DAPS can solve complex, high-dimensional control tasks beyond the pixel-based domain. In particular, we show that the latent space learned by DAPS can be effectively leveraged to train whole-body expressive motions in a robotics simulator, conditioned on a behavior context. Empirically, DAPS trains on high-dimensional datasets (e.g., ImageNet) with superior reconstruction quality, while offering (i) explicit entropy/bit-rate control via the entropy regularization parameter $\beta$, (ii) stochastic discrete latents amenable to downstream search, and (iii) stable training behavior.

## 2 PROBLEM SETTING AND RELATED WORK

We seek to maximize the Evidence Lower Bound (ELBO) of the data likelihood $p(\mathbf{x})$, as introduced in the VAE (Kingma & Welling, 2013; Rezende et al., 2014), but using a discrete latent variable. As in the continuous case, the ELBO is given by the likelihood of the reconstruction minus the KL divergence between the approximate latent posterior and latent prior:

$$\mathcal{L}_{\phi,\theta}^{\text{ELBO}} = \mathbb{E}_{\mathbf{x}}\mathbb{E}_{\mathbf{z}|\mathbf{x}}\Big[\log p_\phi(\mathbf{x}|\mathbf{z})\Big] - \beta D_{\text{KL}}\Big(q_\theta(\mathbf{z}|\mathbf{x}) \,||\, p(\mathbf{z})\Big), \tag{1}$$

where the generative model $p_\phi(\mathbf{x}|\mathbf{z})$ and the recognition model $q_\theta(\mathbf{z}|\mathbf{x})$ are neural networks parameterized by $\phi$ and $\theta$, respectively. The expectations are computed by first sampling $\mathbf{x}$ from the dataset $\mathbf{X}$ and then sampling $\mathbf{z}$ from the recognition model $q_\theta(\mathbf{z}|\mathbf{x})$. Higgins et al. (2017) introduced the coefficient $\beta$ as a hyperparameter, generalizing the original VAE which can be recovered for $\beta = 1$.

In the continuous setting, we can choose a Gaussian recognition model and reparameterize $\mathbf{z}$ to estimate the gradient of the expectation with respect to $\theta$ for gradient-based optimization. However, we consider discrete latent embeddings, which we organize as a sequence of $B$ blocks, $\mathbf{z} = (\mathbf{z}_1, \ldots, \mathbf{z}_B)$, with $V$ possible classes per block (also known as the vocab/codebook size). Furthermore, while our method can be applied to general architectures, we explicitly focus on autoregressive encoders

$$q((\mathbf{z}_1, \ldots, \mathbf{z}_B)|x) = \prod_{i=1}^{B} q(\mathbf{z}_i|\mathbf{z}_1, \ldots, \mathbf{z}_{i-1}, \mathbf{x})$$

due to the higher expressiveness compared to approximating the joint distribution as a product of marginals. However, by using discrete latents, exact differentiable reparameterization is not possible, and approximate backpropagation through the autoregressive sampling (backpropagation through time) is challenging because of high gradient sensitivity. We will now discuss related work, outlining existing solutions for discrete latent variables and comparing their applicability to our problem setting.

**Approximate Discrete Reparameterization**. Methods based on approximate reparameterization typically apply the Gumbel-Softmax trick, an approximation of the exact but non-differentiable Gumbel-Max reparameterization, that was originally proposed by Jang et al. (2016) and applied to the binary MNIST dataset. Given $\mathbf{x} \sim p(\mathbf{x})$, let $\boldsymbol{\ell}$ represent the unnormalized class probabilities such that $q_\theta(\mathbf{z}|\mathbf{x}) = \text{softmax}(q_\theta(\boldsymbol{\ell}|\mathbf{x}))$. Reparameterizing the latent variable can be performed by first sampling Gumbel noise: $\mathbf{g} = -\log(-\log(\mathbf{u}+\epsilon)+\epsilon)$, where $\mathbf{u} \sim \text{Uniform}(\mathbf{0}, \mathbf{1})$ and $\epsilon$ is arbitrarily close to zero. Whereas the Gumbel-Max trick would exploit that $z_{\text{hard}} = \arg\max_i \ell_i + g_i$ follows the desired distribution, $z_{\text{hard}} \sim q(z|\mathbf{x})$, the Gumbel-Softmax computes $\mathbf{z}_{\text{soft}} = \text{softmax}((\boldsymbol{\ell}+\mathbf{g})/\tau)$. The temperature $\tau$ is a hyperparameter that needs to be carefully chosen to trade off the accuracy of the approximation and the variance of the gradient. In order to pass hard assigned labels to the VAE decoder, Jang et al. (2016) combine the Gumbel-Softmax reparameterization with a straight-through estimator, by using the exact but non-differentiable Gumbel-Max reparameterization during the forward pass, and the approximate but differentiable Gumbel-Softmax reparameterization during the backward pass, which can be straightforwardly implemented as $\mathbf{z} := \mathbf{z}_{\text{soft}} + \text{sg}(\mathbf{z}_{\text{hard}} - \mathbf{z}_{\text{soft}})$, where

$z_{\text{hard}}$ refers to the one-hot encoding of $z_{\text{hard}}$ and sg prevents the gradient from flowing through its argument. Although the Gumbel-Softmax trick was successfully used by the discrete VAE (dVAE) in DALL-E (Ramesh et al., 2021) to generate high-quality images, its sensitivity to the temperature parameter is a major hurdle when applying it in practice. Furthermore, backpropagating through the soft-assignments increases the memory footprint, and for autoregressive models the backpropagation-through-time can suffer from vanishing or exploding gradients. We will compare to Gumbel-Softmax in our experiments, where we also consider GR-MCK (Paulus et al., 2020), a modification that uses Rao-Blackwellization for variance reduction.

**Vector Quantization.** VAEs based on vector quantization are arguably the most popular models for learning autoencoders with a discrete latent bottleneck. The VQ-VAE encoder first outputs continuous latent vectors and then maps them to their nearest (in terms of Euclidean distance) vector in a parameterized embedding table $e_\psi$. This quantization discretizes the bottleneck but introduces a non-differentiable argument minimization in the computational graph. To update the encoder with respect to the reconstruction loss, VQ-VAEs employ straight-through estimation, in a similar fashion to the Gumbel-Softmax estimator. Namely, it uses the non-differentiable argument minimization during the forward pass, but skips the quantization step during the backward pass, which can be implemented using a stop-gradient operator as $\mathbf{z}_q := \mathbf{z} + \text{sg}(\mathbf{z}_q - \mathbf{z})$, where $\mathbf{z}$ and $\mathbf{z}_q$ are the encoder outputs before and after quantization, respectively. However, we cannot compute the ELBO in Eq. 1 since we cannot obtain a tractable discrete distribution over the latent variable. Instead, the VQ-VAE optimizes a different loss, given by

$$\mathcal{L}_{\boldsymbol{\phi},\boldsymbol{\theta},\boldsymbol{\psi}} = \mathbb{E}_{\mathbf{x}}\mathbb{E}_{\mathbf{z}|\mathbf{x}}\Big[ -\log p_{\boldsymbol{\phi}}(\mathbf{x}|\mathbf{z}_q) + \|\text{sg}(\mathbf{z}) - e_{\boldsymbol{\psi}}(\mathbf{z})\|_2^2 + \beta\|\mathbf{z} - \text{sg}(e_{\boldsymbol{\psi}}(\mathbf{z}))\|_2^2\Big], \qquad (2)$$

where $e_\psi(\mathbf{z})$ is the nearest embedding vector and $\mathbf{z}_q := \mathbf{z} + \text{sg}(e_\psi(\mathbf{z}) - \mathbf{z})$ uses straight-through estimation. The expectation with respect to $\mathbf{z} \sim q_\theta(\mathbf{z}|\mathbf{x})$ equates to a single sample, i.e., the output of the deterministic recognition model. A more recent method, FSQ Mentzer et al. (2023), replaces the learned codebook with a rounding mechanism. This design choice results in increased parameter efficiency compared to VQ-VAE (due to the small number of levels typically used), while also avoiding the tricks needed to enforce high codebook usage. Similar to the VAE, we focus on generative modeling from compact latent spaces, but in contrast to quantization-based methods, we explicitly regularize for entropy, as staying close to the prior distribution requires more direct control of the encoder entropy. Finally, it is worth noting that the use of heterogeneous FSQ levels can make FSQ more challenging to implement in downstream tasks compared to other discrete VAEs.

**Gradient-Free Optimization.** Another branch of methods circumvents the challenges of backpropagating through the discrete sampling process by relying on zero-order methods. Episode-based policy search (Deisenroth et al., 2013) is a subfield of reinforcement learning which aims to optimize a policy $\pi_\theta(\mathbf{a}|\mathbf{s})$, which is a probability distribution over actions $\mathbf{a}$ given the state $\mathbf{s}$, to maximize the expected return $R$, which is the cumulative reward received over an episode. REINFORCE (Williams, 1992) is a method that can be used to obtain an unbiased estimate of the gradient of the expected return $R$ with respect to the policy parameters $\boldsymbol{\theta}$, without requiring backpropagation through an expectation over the policy. Due to the close relation between reinforcement learning and variational inference (Neumann, 2011; Arenz et al., 2018; Levine, 2018), this approach can be straightforwardly applied to optimize the ELBO 1, where the recognition model $q_\theta(\mathbf{z}|\mathbf{x})$ takes the role of the policy, and the log-likelihood $\log p_\phi(\mathbf{x}|\mathbf{z})$ the role of the return (Tucker et al., 2017; Mnih & Gregor, 2014; Gu et al., 2015; Grathwohl et al., 2017). As REINFORCE suffers from high variance, these methods typically employ variance reduction techniques, such as introducing control variates (Mnih & Gregor, 2014) or combinations of gradient estimators (Tucker et al., 2017; Grathwohl et al., 2017).

However, despite these efforts, the gradient variance of these zero-order methods is still large, resulting in limited success compared to approximate reparameterization or vector quantization. Yet, there have been significant advances in the field of policy search since the introduction of REINFORCE (Williams, 1992) that have been overlooked in the field of discrete VAEs. In particular, trust-regions based on the Kullback-Leibler divergence to the previous policy (Peters et al., 2010; Schulman, 2015) and zero-order natural gradient estimates (Kakade, 2001; Peters & Schaal, 2008; Wierstra et al., 2014) are standard techniques in reinforcement learning that also seem promising for training discrete VAEs. REPS (Peters et al., 2010) combines both ideas by iteratively solving the

optimization problem of maximizing the expected return of a non-parameteric policy subject to a trust-region constraint to the previous policy. REPS has been adapted to many problems, including hierarchical control (Daniel et al., 2016), model-based RL (Abdolmaleki et al., 2015), deep RL (Abdolmaleki et al., 2018), and variational inference (Arenz et al., 2018). Methods that are similar in nature to ours include On-Policy Maximum a Posteriori Optimization (V-MPO) (Song et al., 2019) and Supervised Policy Update (SPU) (Vuong et al., 2018). As in REPS, these methods construct a nonparametric target policy by solving a constrained optimization problem and then update the parameterized policy by minimizing its KL divergence from the target policy, resulting in a weighted maximum likelihood objective. Similar to V-MPO and SPU, we use a neural network-based policy. We also incorporate similar techniques from Lower Bound Policy Search (LBPS) (Watson & Peters, 2023), which optimizes the effective sample size (ESS) to adapt the size of the trust region.

# 3 DISCRETE AUTOENCODING VIA POLICY SEARCH (DAPS)

We will now present our method for optimizing the ELBO (Eq. 1) with respect to expressive discrete encoders, such as transformers, that can capture the correlations between different latent dimensions through auto-regressive sampling. We avoid computationally expensive and high-variance back-propagation through time by using insights from reinforcement learning that enable us to optimize the encoder using weighted maximum likelihood. In the following, we will present the encoder update and the decoder update separately, although they are updated together during training.

Introducing the return $R(\mathbf{z}, \mathbf{x}) = \log p_\phi(\mathbf{x}|\mathbf{z}) + \beta \log p(\mathbf{z})$ we can see that the ELBO (Eq. 1) can be treated as an entropy regularized return maximization problem,

$$J(q) = \int_{\mathbf{x}} p(\mathbf{x}) \sum_{\mathbf{z}} q_\theta(\mathbf{z}|\mathbf{x}) R(\mathbf{z}, \mathbf{x}) \, d\mathbf{x} + \beta H(q(\mathbf{z}|\mathbf{x})), \tag{3}$$

which is closely related to maximum entropy reinforcement learning (Levine, 2018). However, standard reinforcement learning typically considers sequential decision making, where an agent observes the current state $\mathbf{x}$, chooses an action $\mathbf{z}$ according to its policy $q_\theta(\mathbf{z}|\mathbf{x})$, which causes a transition to the next state and yields a step-based reward, and aims to learn the policy that maximizes the accumulated reward. As we are not considering such sequential decision making in our setting, our work is more closely related to episode-based policy search (Deisenroth et al., 2013), a subfield of reinforcement learning that directly aims to maximize the policy with respect to the return, without considering the step-based interactions.

Building on this connection, we can apply techniques from reinforcement learning and policy search to update the encoder of our discrete autoencoder. Namely, we estimate the state value $V(\mathbf{x})$, which corresponds to the expected return and entropy for state $\mathbf{x}$, and use it for variance reduction. Furthermore, we split the encoder updates into two steps, where we first perform a closed-form update on a non-parametric encoder $q(\mathbf{z}|\mathbf{x})$ and then realize this update on the parametric encoder $q_\theta(\mathbf{z}|\mathbf{x})$ using maximum likelihood. While these techniques were introduced in reinforcement learning, and we sometimes use RL terminology to highlight the connections, the following description of our method is self-contained and does not assume prior knowledge in reinforcement learning.

**Baselines for Variance Reduction**. When using Eq. 3 to optimize the ELBO with respect to the encoder, subtracting a state-dependent baseline $b(\mathbf{x})$ from the return $R(\mathbf{z}, \mathbf{x})$ does not affect the optimal solution, and can reduce the variance of Monte-Carlo estimates without introducing a bias (Greensmith et al., 2004; Deisenroth et al., 2013). A typical baseline in reinforcement learning is to use the (soft-)Value function $V(\mathbf{x})$, which corresponds to the expected entropy-regularized return. However, we want to estimate a suitable baseline using a small number $k$ of latent samples, and, therefore, estimating the expected return would be highly sensitive to bad outliers. Instead, we use the soft-maximum of the returns, which is an optimistic estimate of the soft-Value function, and accurate under the assumption that the samples were drawn from the optimal encoder (Ziebart, 2010). Hence, we replace the return with the estimated advantage

$$A(\mathbf{z}, \mathbf{x}) = R(\mathbf{z}, \mathbf{x}) - \log \sum_{k=1}^{K} \exp R(\mathbf{z}^k, \mathbf{x}), \quad \mathbf{z}^k \sim q_\theta(\mathbf{z}|\mathbf{x}).$$

**Optimizing the Nonparametric Encoder**. We will start by defining a constrained optimization problem for the objective given by Eq. 3. The procedure is based on REPS, where the objective is

---

**Algorithm 1** DAPS: Weight Computation and Loss Functions

**Procedure** GETWEIGHTS$(\boldsymbol{\theta}, \boldsymbol{\phi}, \eta, \mathbf{x})$

1: **for** $k = 1$ to $K$ **do**
2: $\quad \mathbf{z}^k \leftarrow$ sample from $q_{\boldsymbol{\theta}}(.\,|\mathbf{x})$
3: $\quad \mathcal{R}^k \leftarrow \log p_{\boldsymbol{\phi}}(\mathbf{x}|\mathbf{z}^k) + c$
4: $\mathbf{A}(\mathbf{z}, \mathbf{x}) \leftarrow \mathcal{R} - \log \sum_k \exp(\mathcal{R}^k)$
5: $\log q^*(\mathbf{z}|\mathbf{x}) \propto \dfrac{\mathbf{A}(\mathbf{z}, \mathbf{x}) + \eta \log q_{\boldsymbol{\theta}}(\mathbf{z}|\mathbf{x})}{\eta + \beta}$
6: $w^k \leftarrow \dfrac{q^*(\mathbf{z}^k|\mathbf{x})}{q_{\boldsymbol{\theta}}(\mathbf{z}^k|\mathbf{x})} \left( \sum_k \dfrac{q^*(\mathbf{z}^k|\mathbf{x})}{q_{\boldsymbol{\theta}}(\mathbf{z}^k|\mathbf{x})} \right)^{-1}$
7: **return** stopGrad$(\{w^k\}, \{\mathbf{z}^k\})$

**Procedure** UPDATEFN$(\boldsymbol{\theta}, \boldsymbol{\phi}, \eta, \mathbf{x})$

1: $\{w^k\}, \{\mathbf{z}^k\} \leftarrow$ GETWEIGHTS$(\boldsymbol{\theta}, \boldsymbol{\phi}, \eta, \mathbf{x})$
2: **// Decoder update**
3: $\mathcal{L}(\boldsymbol{\phi}) \leftarrow -\frac{1}{K} \sum_k \log p_{\boldsymbol{\phi}}(\mathbf{x}|\mathbf{z}^k)$
4: **// Encoder update**
5: $\mathcal{L}(\boldsymbol{\theta}) \leftarrow -\sum_k w^k \log q_{\boldsymbol{\theta}}(\mathbf{z}^k|\mathbf{x})$
6: **// update for $\eta$**
7: $\mathcal{L}(\eta) \leftarrow \left( \widehat{\mathrm{ESS}}_\eta(q^*, q_\theta) - \mathrm{ESS}_{\mathrm{target}} \right)^2$
8: **return** $\mathcal{L}(\boldsymbol{\theta}), \mathcal{L}(\boldsymbol{\phi}), \mathcal{L}(\eta)$

---

to maximize the expected return (or advantage) while satisfying a constraint on the KL divergence between the policy and the most recent parametric policy $q_\theta(\mathbf{z}|\mathbf{x})$,

$$\max_q \quad \int_{\mathbf{x}} p(\mathbf{x}) \sum_{\mathbf{z}} q(\mathbf{z}|\mathbf{x}) A(\mathbf{z}, \mathbf{x}) \, dx + \beta H(q(\mathbf{z}|\mathbf{x}))$$
$$\text{s.t.} \quad D_{\mathrm{KL}}(q(\mathbf{z}|\mathbf{x}) \,\|\, q_{\boldsymbol{\theta}}(\mathbf{z}|\mathbf{x})) \leq \epsilon_\eta \quad , \quad \sum_{\mathbf{z}} q(\mathbf{z}|\mathbf{x}) = 1. \tag{4}$$

We can use Lagrangian multipliers to convert the constrained optimization problem into an unconstrained one. As shown in Appendix A.1, the optimal solution is given by

$$q^*(\mathbf{z}|\mathbf{x}) \propto \exp \left( \frac{A(\mathbf{z}, \mathbf{x}) + \eta \log q_{\boldsymbol{\theta}}(\mathbf{z}|\mathbf{x})}{\eta + \beta} \right), \tag{5}$$

where $\eta$ is a Lagrangian multiplier. The term $\eta$ controls the trust region, i.e., how large of a step we take toward the optimal policy, while $\beta$ controls the policy entropy. While Eq. 5 provides us with a solution for the optimal policy, we can only evaluate it on given particles, and we cannot compute the normalizer, since summing over all actions is intractable when using a high-dimensional latent bottleneck. However, we will now show that we can use Eq. 5 to evaluate importance weights to update the parametric policy using weighted maximum likelihood.

**Updating the Parametric Encoder**. Now that we have the form of the optimal nonparametric policy, we can move our parameterized policy toward it using maximum likelihood, which enables us to train it without requiring backpropagation through the autoregressive sampling. However, because we cannot sample from our nonparametric distribution $q^*$, we have to resort to importance sampling, where we use the current parametric encoder as the proposal distribution. Hence, we maximize a weighted maximum likelihood objective,

$$\mathcal{L}(\boldsymbol{\theta}) = \int_{\mathbf{x}} p(\mathbf{x}) \, D_{\mathrm{KL}}(q^*(\mathbf{z}|\mathbf{x}) \,\|\, q_{\boldsymbol{\theta}}(\mathbf{z}|\mathbf{x})) \, d\mathbf{x} \approx -\frac{1}{N} \sum_{i=1}^{N} \sum_{k=1}^{K} w_{ik} \log q_{\boldsymbol{\theta}}(\mathbf{z}^k|\mathbf{x}_i) \, + \, \mathrm{const} \tag{6}$$

where $w_i = q^*(\mathbf{z}|\mathbf{x}_i) q_\theta(\mathbf{z}|\mathbf{x}_i)^{-1}$ are the importance weights, and the constant term is the entropy of $q^*$ which does not affect optimization. However, due to the unknown normalizer of $q^*$, we can only evaluate the importance weights up to a constant factor, and therefore resort to self-normalized importance weighting using weights $\tilde{w}_i = w_i / \sum_j w_j$. Self-normalized importance weights benefit from a lower variance but introduce a bias to the approximation, which diminishes asymptotically for large sample sizes. The new objective can then be summarized as maximizing the log-likelihood of our parameterized policy, weighted by $q^*$. The generative model is fixed during the policy update, and the process for updating it will now be described in more detail.

**Decoder Update**. The generative model and the recognition model (i.e., the policy) are updated using coordinate descent, which separates the optimization of the decoder from the optimization of the encoder. The generative model $p_{\boldsymbol{\phi}}(\mathbf{x}|\mathbf{z})$ takes in a latent code $\mathbf{z}$ and outputs the parameters of a distribution over $\mathbf{x}$. When optimizing the ELBO with respect to $\phi$, the objective reduces to maximizing the expected log-likelihood under the current recognition model:

$$\mathcal{L}(\boldsymbol{\phi}) = -\int_{\mathbf{x}} p(\mathbf{x}) \sum_{\mathbf{z}} q_{\boldsymbol{\theta}}(\mathbf{z}|\mathbf{x}) \log p_{\boldsymbol{\phi}}(\mathbf{x}|\mathbf{z}) \, d\mathbf{x} \approx -\sum_i \mathbb{E}_{\mathbf{z} \sim q_{\boldsymbol{\theta}}(\mathbf{z}|\mathbf{x}_i)} \left[ \log p_{\boldsymbol{\phi}}(\mathbf{x}_i|\mathbf{z}) \right]. \tag{7}$$

The approximation follows from Monte Carlo integration using randomly sampled data points and latent samples. Note that this objective corresponds to the standard loss of the VAE in Eq. 1. However, because we do not reparameterize the latent variable, the update is performed independently from the policy update, in contrast to VAEs, where both the generative model and recognition model are updated in one backward pass. Hence, our generative model update does not depend on $\boldsymbol{\theta}$, and the KL divergence to the latent prior can be dropped.

**Step Size Adaptation using Effective Sample Size**. The multiplier $\eta$ in Eq. 5 controls the step size of our update based on the relative entropy between our nonparametric target $q^*$ and $q_\theta$. To adapt $\eta$ automatically, we follow prior work using the *effective sample size* (ESS) (Maia Polo & Vicente, 2023; Metelli et al., 2020; Watson & Peters, 2023), which provides a tractable proxy for the order-2 Rényi divergence. We update $\eta$ so that the minibatch ESS matches a desired target level $\text{ESS}_{\text{target}}$:

$$\widehat{\text{ESS}}_\eta = \frac{1}{N} \sum_{i=1}^{N} \frac{\left(\sum_{k=1}^{K} w_{ik}\right)^2}{\sum_{k=1}^{K} w_{ik}^2}, \qquad w_{ik} = \frac{q^*(z_i^k \mid x_i; \eta)}{q_\theta(z_i^k \mid x_i)}. \tag{8}$$

Further details and the connection to Rényi divergences are provided in Appendix A.2. In practice, $\eta$ is treated as a trainable parameter and updated with SGD on $(\widehat{\text{ESS}}_\eta - \text{ESS}_{\text{target}})^2$. We found $\text{ESS}_{\text{target}} \in [K/4, 3K/4]$ to yield stable convergence. During training, we observe that $\eta$ decays smoothly over time, resulting in smoothly decreasing step sizes during optimization, while accounting for the task-specific scale of the reconstruction loss.

## 4 EXPERIMENTS

**Datasets.** We evaluate on four domains of increasing scale and complexity. We will use the term *block size* to denote the number of indices in the latent code (i.e., the maximum length of the transformer decoder), and the term *vocab size* to denote the number of possible values each index can take (i.e., the number of embeddings in the VQ-VAE codebook). The datasets are as follows. **MNIST**: $28 \times 28$ (binary) with a 64-bit bottleneck (vocab size 256; block size 8). **CIFAR-10**: $32 \times 32$ with a 576-bit bottleneck (vocab size 512; block size 64). **ImageNet-256**: $256 \times 256$ with a 10,240-bit bottleneck (vocab size 1,024; block size 1,024). **LAFAN**: motion dataset ($\sim$4.6h expressive human motion, 5 subjects) with a 640-bit bottleneck (vocab size 1,024; block size 64). For LAFAN, motions are retargeted to the Unitree H1 robot and encoded as $32 \times 121$ tensors of relative poses. Across datasets, we enforce the same bottleneck size for all methods to ensure fair comparison.

**Architectures.** We adopt an MLP generative model for MNIST (inspired by (Kingma & Welling, 2013)), and a ResNet decoder for CIFAR/ImageNet (with additionally a learnable variance clipped to $[0.01, 1.0]$) (Van Den Oord et al., 2017). The recognition model is a vision transformer (Alexey, 2020), enabling either autoregressive sampling (non-VQ methods) or direct encoding of codebook vectors (VQ methods) as depicted in Appendix I. For completeness, we also tested the ResNet encoder from (Van Den Oord et al., 2017) in the VQ setting, but observed no advantage. For LAFAN, both recognition and generative models are transformer-based (See Appendix J). Because VQ-based methods rely on a decoder-only transformer while ELBO-based methods use an encoder–decoder architecture, we increase the number of layers and hidden units in the VQ decoders to match their total parameter count as closely as possible to ours.

**Training.** All methods are trained with identical batch sizes and numbers of optimizer steps per dataset. We use Adam with weight decay and a base learning rate of $3 \times 10^{-4}$, applying cosine decay when appropriate. For MNIST and CIFAR, we train for 250k steps with a batch size of 256 on an RTX-3090. For LAFAN, we also train for 250k steps with a batch size of 256 on an A100-40GB. For ImageNet, we run for 300k steps with a batch size of 64 on A100-80GB GPUs (see Appendix H). Each experiment is repeated with 10 random seeds on MNIST, CIFAR, and LAFAN, and with 5 seeds on ImageNet due to the higher cost. In practice, we find that the autoregressive Gumbel Softmax is unstable at the base learning rate, requiring per-dataset tuning. Because of this instability, we exclude it from ImageNet experiments. In contrast, autoregressive GR-MCK remains stable in the ImageNet setting, though it requires more GPU memory.

**Baselines.** We benchmark DAPS against a range of discrete latent variable models: (1) Gumbel-Softmax, (2) GR-MCK, (3) FSQ, and (4) VQ-VAE. For DAPS, Gumbel Softmax, and GR-MCK, we

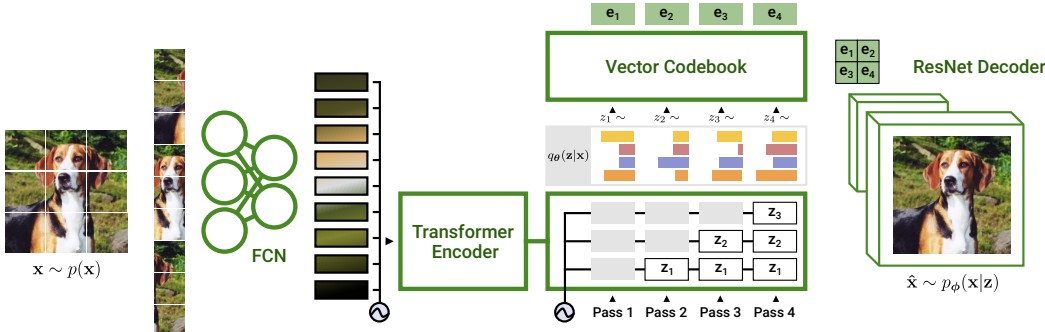

Figure 1: Overview of DAPS. **Left:** Images are split into patches and embedded by a feed-forward network before entering the encoder. **Middle:** The decoder generates latent sequences autoregressively: each step conditions on the encoder output and previously sampled latent codes (via causal masking, shown in gray). **Right:** The generative model decodes latent embeddings into an image.

also implement non-autoregressive versions (denoted by the "NA" suffix), where the decoder-only transformer outputs all logits in a single forward pass. Despite being less expressive, Gumbel-NA outperforms Gumbel on most datasets, which we attribute to the difficulty of gradient estimation in autoregressive sampling. For completeness, we also evaluate the Gaussian VAE (equipped with the same latent capacity) and a PPO-style autoencoding baseline; however, we omit them from larger-scale experiments due to their substantially weaker performance. For each method, we perform a Bayesian sweep over hyperparameters (see Appendix I, J), centered around the default values used in the original works.

**Ablations.** We study the two main hyperparameters of DAPS—$\beta$ and the ESS target—to evaluate the method's sensitivity to their values. Specifically, we consider a grid with $\beta \in \{0.1, 1.0, 5.0, 10.0\}$ and ESS target $\in \{K/4, K/2, 3K/4\}$ on CIFAR-10. Our results show that annealing $\beta$ yields the strongest performance (see Appendix C). Intuitively, this schedule promotes exploration early in training and gradually shifts the focus toward high-quality reconstructions later on. In contrast, DAPS shows little sensitivity to the choice of ESS target. This parameter primarily influences the adaptation of $\eta$ during training to satisfy the entropy objective. Overall, these findings highlight that $\beta$ is the more critical hyperparameter for performance, while the ESS target has a comparatively minor effect, underscoring the robustness of DAPS to its setting. Finally we study the utilization of the latent vocabulary (see Appendix G). The marginal distribution of $q(\mathbf{z}_i|\mathbf{x})$ is displayed in Fig. 2, indicating strong codebook utilization for DAPS compared to FSQ and VQ-VAE.

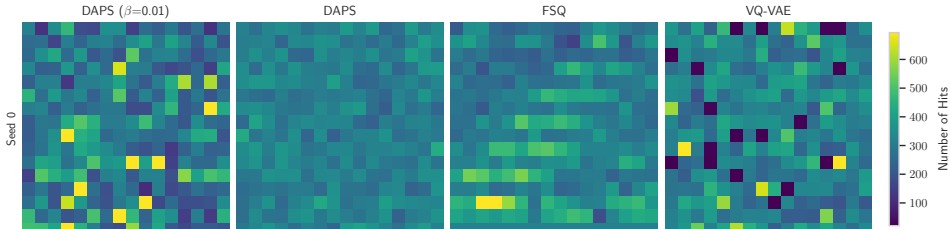

Figure 2: Latent Code Utilization on MNIST Validation Dataset.

## 5 RESULTS

We now turn to an empirical evaluation of DAPS. We present key metrics, including the PSNR (peak signal-to-noise ratio), the $\beta$-ELBO (the $\beta$-weighted latent prior KL divergence plus the log likelihood of the reconstruction), and the FID score (computed using all $50k$ validation samples for ImageNet and all 10k validation samples for CIFAR-10).

| Method | MNIST (64 bits) | | CIFAR-10 (576 bits) | | | ImageNet (1.28 KB) | | | LAFAN (640 bits) | |
|---|---|---|---|---|---|---|---|---|---|---|
| | $\beta$-ELBO | PSNR | $\beta$-ELBO | PSNR | FID | $\beta$-ELBO | PSNR | FID | $\beta$-ELBO | PSNR |
| FSQ | – | 18.42 | – | 24.19 | 163.00 | – | 24.24 | 54.54 | – | 36.19 |
| VQVAE | – | **18.45** | – | 24.19 | 164.30 | – | 23.83 | 65.01 | – | 31.04 |
| DAPS | **-46.54** | 18.23 | **1185.51** | **25.21** | 157.27 | **87.0k** | **24.66** | **48.65** | **-949.78** | **36.81** |
| DAPS-NA | -46.96 | 18.36 | 977.39 | 25.02 | **156.33** | 78.8k | 24.40 | 57.43 | -1050.25 | 32.90 |
| GRMCK | -62.25 | 16.78 | 217.45 | 22.69 | 179.88 | 60.7k | 23.01 | 73.21 | -1008.12 | 34.11 |
| GRMCK-NA | -68.52 | 16.22 | 61.96 | 22.71 | 172.65 | 21.4k | 21.85 | 99.15 | -1530.51 | 26.93 |
| Gumbel | -68.07 | 16.30 | 704.92 | 23.74 | 169.87 | – | – | – | -998.49 | 34.51 |
| Gumbel-NA | -47.09 | 18.21 | 785.35 | 24.27 | 162.04 | 85.2k | 24.49 | 51.66 | -1400.05 | 27.89 |
| PPO | -104.66 | 14.56 | 416.31 | 23.30 | 176.29 | – | – | – | – | – |

Table 1: Summary of mean validation metrics (for best value obtained per seed) across seeds.

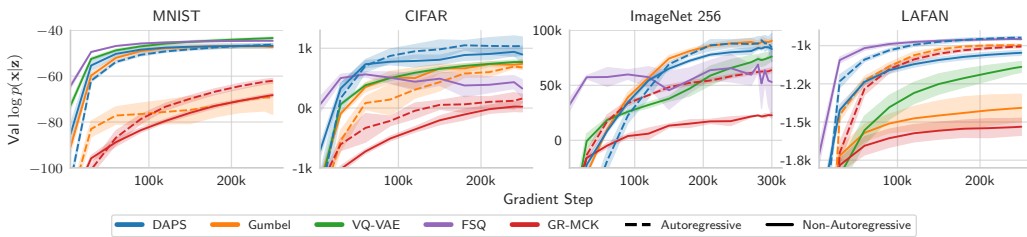

Figure 3: Validation reconstruction log-likelihoods, $\log p(\mathbf{x}|\mathbf{z})$, throughout training ($\beta = 0.01$).

To facilitate better comparison between ELBO-based methods (which incorporate additional entropy regularization) and VQ-based methods, we present our results in two separate tables. In Table 1, the final value of $\beta$ is treated as a hyperparameter rather than as part of the objective. As a result, ELBO-based methods use a final $\beta$ of $0.01$ for MNIST, CIFAR, and LAFAN in Table 1. Meanwhile, a final $\beta$ of $1.0$ is used for ImageNet, which we find works well due to the large magnitude of the reconstruction log-likelihood relative to the latent prior KL divergence. We also seek to compare methods with the original ELBO objective in mind. Consequently, Table 2 presents a comparison of ELBO-based methods for $\beta = 1$. In both settings, DAPS exhibits strong performance, demonstrating the flexibility of our method in accommodating various objectives.

**MNIST.** DAPS achieves comparable performance to baseline methods in terms of $\beta$-ELBO and PSNR on MNIST. However, unlike VQ-based methods, DAPS also produces visually high-quality digits when decoding samples from the latent prior (see Appendix, Fig. 21). We also train a label-conditioned model on the learned latent space, the results for which are shown in Appendix Fig. 20.
**CIFAR-10.** Our method outperforms all baselines on CIFAR-10 across the relevant metrics. Notably, non-autoregressive DAPS (DAPS-NA) achieves a slightly lower FID than DAPS, though it underperforms DAPS on other experiments. Compared to non-VQ-based approaches, DAPS reaches a high ELBO quickly and stably, without the hyperparameter sensitivity observed in methods such as autoregressive Gumbel Softmax (e.g., temperature tuning).

| Method ($\beta = 1$) | MNIST (64 bits) | | CIFAR-10 (576 bits) | | |
|---|---|---|---|---|---|
| | $\beta$-ELBO | PSNR | $\beta$-ELBO | PSNR | FID |
| DAPS | **-80.64** | 17.13 | **857.77** | **25.27** | 158.61 |
| DAPS-NA | -86.00 | 16.54 | 585.02 | 24.97 | **157.16** |
| GRMCK | -101.86 | 14.83 | -211.33 | 21.98 | 184.65 |
| GRMCK-NA | -114.61 | 14.21 | -780.04 | 20.90 | 183.46 |
| Gumbel | -93.72 | 15.68 | 405.18 | 23.51 | 172.68 |
| Gumbel-NA | -80.80 | **17.21** | 394.19 | 24.20 | 163.98 |

Table 2: Mean validation metrics for $\beta = 1.0$ on MNIST and CIFAR-10.

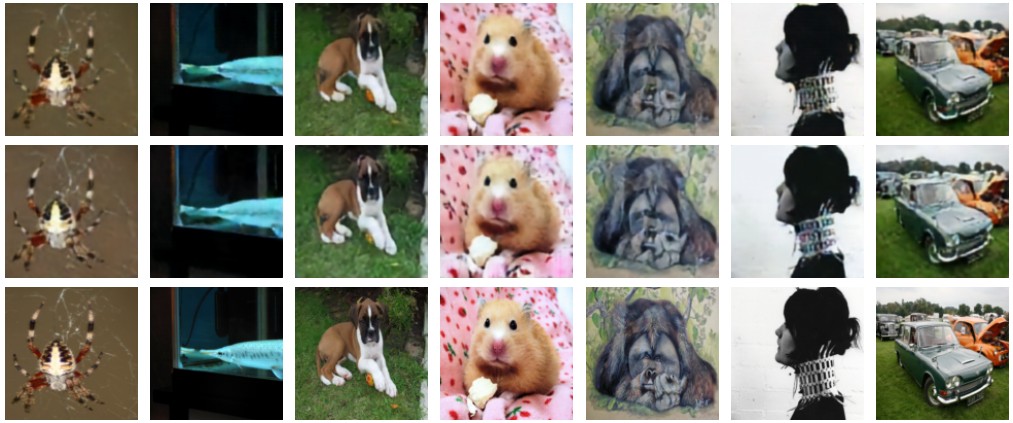

Figure 4: ImageNet 256 validation reconstructions. Top to bottom: DAPS, FSQ, Groundtruth.

**ImageNet.** We evaluate DAPS on the ImageNet-256 dataset, demonstrating its ability to reconstruct high-dimensional data using a low-bit bottleneck. DAPS significantly outperforms traditional methods in terms of reconstruction quality while maintaining high bit efficiency. Given randomly-sampled images from the validation set, DAPS produces high-quality reconstructions, as shown in Fig. 4 and Fig. 23. Finally, DAPS achieves the best $\beta$-ELBO, PSNR, and FID scores comparatively.

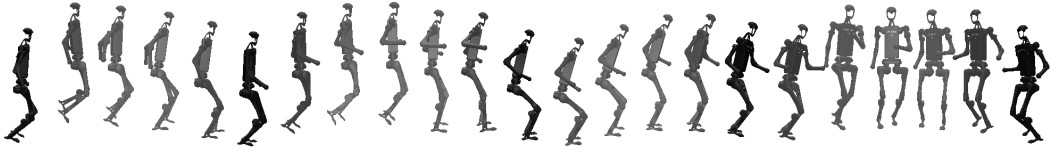

Figure 5: A latent sequence for hopping is decoded by DAPS and fed through inverse kinematics.

**LAFAN.** We also evaluate DAPS on a LAFAN dataset for the Unitree H1 robot provided in LocoMuJoCo (Al-Hafez et al., 2023), which contains human-to-robot retargeted motion trajectories with inherent temporal dependencies and multimodal characteristics. Quantitatively, DAPS excels at reconstructing this data, outperforming baseline methods in terms of $\beta$-ELBO and PSNR. DAPS generates coherent and high-quality motion trajectories that align well with the true behaviors in LAFAN (see the supplementary videos). This makes DAPS amenable to downstream robotics tasks such as hierarchical control, where a high-level policy can guide a low-level policy by specifying a desired trajectory, given a context. A reconstructed motion is shown in Figure 5.

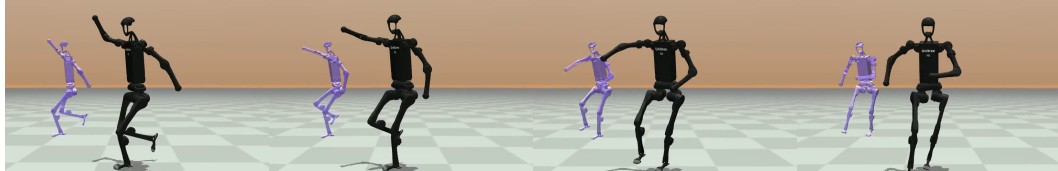

Figure 6: A latent space sequence for dancing is decoded by a reinforcement learning policy and executed in a physics simulator.

**Downstream Task: Goal-Conditioned Robot Control.**

We construct a dataset of motion segments paired with contextual signals and targets. A high-level policy is trained to process a history of relative body poses while conditioning on two context signals: (i) a desired future COM velocity and (ii) a text description of the motion. The policy is trained to autoregressively generate discrete latents in the DAPS latent space in a distillation setup similar to PoseGPT (Lucas et al., 2022). These latents effectively replace raw motion references, providing a command space—defined by text and COM trajectories—that guides a low-level motion imitation policy (see Appendix K). Supplementary videos illustrate the resulting robot behaviors and highlight the utility of DAPS in downstream control tasks. An example motion is shown in Figure 6.

# 6 DISCUSSION

Overall, DAPS provides state-of-the-art performance in reconstructing high-dimensional datasets, both for images and sequential motion data, outperforming existing methods across key metrics. The scalability of DAPS, especially with respect to high-dimensional and sequential data, makes it a strong candidate for a variety of generative tasks. Our approach to discrete variational autoencoding via policy search offers a viable alternative to traditional methods like Gumbel-Softmax and VQ-VAE, particularly when dealing with high-dimensional datasets. Using a standard set of VAE hyperparameters, the main parameters for DAPS are the desired effective sample size and $\beta$. This makes the training process comparable to that of VQ-VAE, which requires the specification of a commitment coefficient. Both methods introduce hyperparameters that are not highly sensitive, making them relatively easy to tune. While autoregressive discrete sampling allows for expressive multimodal latent generation, it introduces a computational bottleneck, particularly as the sequence length increases. This limits the scalability of both DAPS and Gumbel-Softmax when compared to traditional VAEs and VQ-VAEs. However, DAPS avoids backpropagating through autoregressive sampling, allowing it to scale to sequence lengths of 1,024 and likely beyond. This provides an advantage in terms of computational efficiency and scalability within the class of autoregressive methods, particularly on large datasets such as ImageNet 256. Our formulation also allows for fine-grained control over the tradeoff between compression and generalization through the use of $\beta$ in the ELBO. By adjusting $\beta$, we can effectively manage the tradeoff between generating high-quality reconstructions and generalization. This flexibility is a key strength of DAPS over other methods, such as FSQ and VQ-VAE, which do not allow for easy control of entropy. Furthermore, our approach demonstrates stable training across seeds, which is critical for real-world applications.

# 7 CONCLUSION

We introduced DAPS, a policy-search-based framework for training discrete variational autoencoders without relying on reparameterization techniques. Our method integrates reinforcement learning concepts to optimize the variational encoder using weighted maximum likelihood, providing both scalability and efficiency. Experiments on high-dimensional datasets such as ImageNet and LAFAN demonstrate DAPS' superior performance in reconstruction quality and its ability to generalize well across different tasks. By combining insights from policy search with variational autoencoding, DAPS opens the door to the efficient use of discrete latent bottlenecks for large-scale generative modeling. Its flexibility in balancing compression and generalization, along with its ability to handle complex data like images and motion trajectories, positions DAPS as a powerful tool for both generative modeling and downstream control tasks. This work also paves the way for further exploration of policy-driven techniques in the optimization of generative models.

ACKNOWLEDGMENTS

This work was supported by the German Science Foundation (DFG) under project RTG2761. We gratefully acknowledge support from the hessian.AI Service Center (funded by the Federal Ministry of Education and Research, BMBF, grant no. 01IS22091), the hessian.AI Innovation Lab (funded by the Hessian Ministry for Digital Strategy and Innovation, grant no. S-DIW04/0013/003), and the Lichtenberg high-performance computer of TU Darmstadt. This work was also supported by a hardware donation from NVIDIA through the Academic Grant Program.

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

# APPENDIX

## A DAPS DERIVATIONS

### A.1 SOLVING THE CONSTRAINED OPTIMIZATION PROBLEM

We wish to solve the following constrained optimization problem:

$$
\begin{aligned}
\max_{q} \quad & \int_{\mathbf{x}} p(\mathbf{x}) \sum_{\mathbf{z}} q(\mathbf{z}|\mathbf{x}) A(\mathbf{z}, \mathbf{x}) \, dx + \beta \mathcal{H}(q(\mathbf{z}|\mathbf{x})) \\
\text{s.t.} \quad & D_{\mathrm{KL}}(q(\mathbf{z}|\mathbf{x}) \parallel q_{\boldsymbol{\theta}}(\mathbf{z}|\mathbf{x})) \leq \epsilon_{\eta}, \quad \sum_{\mathbf{z}} q(\mathbf{z}|\mathbf{x}) = 1
\end{aligned}
\tag{9}
$$

Using Lagrangian multipliers, we can convert this into an unconstrained optimization problem:

$$
\begin{aligned}
\mathcal{J}(q(\mathbf{z}|\mathbf{x}), \eta, \lambda) = & \int_{\mathbf{x}} p(\mathbf{x}) \sum_{o} q(\mathbf{z}|\mathbf{x}) A(\mathbf{x}, \mathbf{z}) d\mathbf{x} + \beta \mathcal{H}(q(\mathbf{z}|\mathbf{x})) \\
& + \eta \Big( \epsilon_{\eta} - \int_{\mathbf{x}} p(\mathbf{x}) \sum_{\mathbf{z}} q(\mathbf{z}|\mathbf{x}) \log \frac{q(\mathbf{z}|\mathbf{x})p(\mathbf{x})}{q_{\boldsymbol{\theta}}(\mathbf{z}|\mathbf{x})p(\mathbf{x})} d\mathbf{x} \Big) \\
& + \int_{\mathbf{x}} p(\mathbf{x}) \Big[ \lambda(\mathbf{x})(1 - \sum_{\mathbf{z}} q(\mathbf{z}|\mathbf{x})) \Big] d\mathbf{x}
\end{aligned}
\tag{10}
$$

Differentiating $\mathcal{J}$ with respect to $q(\mathbf{z}|\mathbf{x})$ and setting to zero yields:

$$
\begin{aligned}
\frac{\delta \mathcal{J}(q, \eta, \lambda)}{\delta q} = & \int_{\mathbf{x}} \sum_{\mathbf{z}} \frac{\delta}{\delta q} \Big[ p(\mathbf{x}) q(\mathbf{z}|\mathbf{x}) \Big( \eta \log q_{\boldsymbol{\theta}}(\mathbf{z}|\mathbf{x}) \\
& - \eta \log q(\mathbf{z}|\mathbf{x}) - \lambda(\mathbf{x}) + A(\mathbf{x}, \mathbf{z}) - \beta \log q(\mathbf{z}|\mathbf{x}) \Big) \Big] d\mathbf{x} \\
0 \stackrel{!}{=} & \, p(\mathbf{x}) \Big( \eta \log q_{\boldsymbol{\theta}}(\mathbf{z}|\mathbf{x}) - \eta \log q(\mathbf{z}|\mathbf{x}) - \eta - \lambda(\mathbf{x}) + A(\mathbf{x}, \mathbf{z}) - \beta \log q(\mathbf{z}|\mathbf{x}) - \beta \Big)
\end{aligned}
\tag{11}
$$

Re-arranging terms gives us the optimal nonparametric target distribution, $q^*(\mathbf{z}|\mathbf{x})$:

$$
\begin{aligned}
q^*(\mathbf{z}|\mathbf{x}) &= \exp \Big( \frac{A(\mathbf{x}, \mathbf{z}) + \eta \log q_{\boldsymbol{\theta}}(\mathbf{z}|\mathbf{x})}{\eta + \beta} \Big) \exp \Big( \frac{\eta + \beta + \lambda(\mathbf{x})}{\eta + \beta} \Big)^{-1} \\
&= q_{\boldsymbol{\theta}}(\mathbf{z}|\mathbf{x}) \exp \Big( \frac{A(\mathbf{x}, \mathbf{z}) - \beta \log q_{\boldsymbol{\theta}}(\mathbf{z}|\mathbf{x})}{\eta + \beta} \Big) \exp \Big( \frac{\eta + \beta + \lambda(\mathbf{x})}{\eta + \beta} \Big)^{-1},
\end{aligned}
\tag{12}
$$

whereby re-writing the last line in Eq. 12, we can see that the optimal policy is a posterior distribution with a prior that is the parametric policy $q_{\boldsymbol{\theta}}(\mathbf{z}|\mathbf{x})$. This gives us the following dual function:

$$
\begin{aligned}
\mathcal{G}(\eta, \lambda(\mathbf{x})) = & \int_{\mathbf{x}} p(\mathbf{x}) \lambda(\mathbf{x}) d\mathbf{x} + \eta \epsilon_{\eta} \\
& + \int_{\mathbf{x}} p(\mathbf{x}) \sum_{\mathbf{z}} q^*(\mathbf{z}|\mathbf{x}) \Big[ \eta \log q_{\boldsymbol{\theta}}(\mathbf{z}|\mathbf{x}) - (\eta + \beta) \log q^*(\mathbf{z}|\mathbf{x}) - \lambda(\mathbf{x}) + A(\mathbf{x}, \mathbf{z}) \Big] d\mathbf{x} \\
= & \int_{\mathbf{x}} p(\mathbf{x}) \lambda(\mathbf{x}) d\mathbf{x} + \eta \epsilon_{\eta} + (\eta + \beta) \int_{\mathbf{x}} p(\mathbf{x}) \sum_{\mathbf{z}} q^*(\mathbf{z}|\mathbf{x}) d\mathbf{x}
\end{aligned}
\tag{13}
$$

Since the optimal $\lambda(\mathbf{x})$ normalizes $q^*(\mathbf{z}|\mathbf{x})$, we have:

$$\exp\left(\frac{\eta + \beta + \lambda^*(\mathbf{x})}{\eta + \beta}\right) = \sum_{\mathbf{z}} \exp\left(\frac{A(\mathbf{x}, \mathbf{z}) + \eta \log q_{\boldsymbol{\theta}}(\mathbf{z}|\mathbf{x})}{\eta + \beta}\right)$$

$$\lambda^*(\mathbf{x}) = (\eta + \beta) \log \sum_{\mathbf{z}} \exp\left(\frac{A(\mathbf{x}, \mathbf{z}) + \eta \log q_{\boldsymbol{\theta}}(\mathbf{z}|\mathbf{x})}{\eta + \beta}\right) - \eta - \beta \quad (14)$$

Plugging this into the dual, we get:

$$\mathcal{G}(\eta, \lambda^*(\mathbf{x})) = \int_{\mathbf{x}} p(\mathbf{x})\Big[(\eta + \beta) \log \sum_{\mathbf{z}} \exp\left(\frac{A(\mathbf{x}, \mathbf{z}) + \eta \log q_{\boldsymbol{\theta}}(\mathbf{z}|\mathbf{x})}{\eta + \beta}\right) - \eta - \beta\Big]d\mathbf{x} + \eta\epsilon_\eta$$

$$+ (\eta + \beta) \int_{\mathbf{x}} p(\mathbf{x}) \sum_{\mathbf{z}} q^*(\mathbf{z}|\mathbf{x})d\mathbf{x} \quad (15)$$

$$= (\eta + \beta) \int_{\mathbf{x}} p(\mathbf{x})\Big[\log \sum_{\mathbf{z}} \exp\left(\frac{A(\mathbf{x}, \mathbf{z}) + \eta \log q_{\boldsymbol{\theta}}(\mathbf{z}|\mathbf{x})}{\eta + \beta}\right)\Big]d\mathbf{x} + \eta\epsilon_\eta$$

We can now differentiate with respect to $\eta$ to optimize our stepsize.

$$\frac{\delta\mathcal{G}(\eta, \lambda^*(\mathbf{x}))}{\delta\eta} = \frac{\delta}{\delta\eta}\Big[(\eta + \beta) \int_{\mathbf{x}} p(\mathbf{x})\Big[\log \sum_{\mathbf{z}} \exp\left(\frac{A(\mathbf{x}, \mathbf{z}) + \eta \log q_{\boldsymbol{\theta}}(\mathbf{z}|\mathbf{x})}{\eta + \beta}\right)\Big]d\mathbf{x} + \eta\epsilon_\eta\Big] \quad (16)$$

Letting $v = \sum_{\mathbf{z}} \exp\left(\frac{A(\mathbf{x}, \mathbf{z}) + \eta \log q_{\boldsymbol{\theta}}(\mathbf{z}|\mathbf{x})}{\eta + \beta}\right)$ and $u = \log v$, we have that $\frac{\delta u}{\delta v} = v^{-1}$ and:

$$\frac{\delta v}{\delta\eta} = \sum_{\mathbf{z}} \left(\frac{\beta \log q_{\boldsymbol{\theta}}(\mathbf{z}|\mathbf{x}) - A(\mathbf{x}, \mathbf{z})}{(\eta + \beta)^2}\right) \exp\left(\frac{A(\mathbf{x}, \mathbf{z}) + \eta \log q_{\boldsymbol{\theta}}(\mathbf{z}|\mathbf{x})}{\eta + \beta}\right) \quad (17)$$

Applying the chain rule, using $\frac{\delta u}{\delta\eta} = \frac{\delta u}{\delta v}\frac{\delta v}{\delta\eta}$, we get:

$$\frac{\delta\mathcal{G}(\eta, \lambda^*(\mathbf{x}))}{\delta\eta} = \int_{\mathbf{x}} p(\mathbf{x})\Big[\sum_{\mathbf{z}} \left(\frac{\beta \log q_{\boldsymbol{\theta}}(\mathbf{z}|\mathbf{x}) - A(\mathbf{x}, \mathbf{z})}{\eta + \beta}\right)\frac{\exp\left(\frac{A(\mathbf{x}, \mathbf{z}) + \eta \log q_{\boldsymbol{\theta}}(\mathbf{z}|\mathbf{x})}{\eta + \beta}\right)}{\sum_{\mathbf{z}'} \exp\left(\frac{A(\mathbf{x}, \mathbf{z}') + \eta \log q_{\boldsymbol{\theta}}(\mathbf{z}'|\mathbf{x})}{\eta + \beta}\right)}$$

$$+ \log \sum_{\mathbf{z}} \exp\left(\frac{A(\mathbf{x}, \mathbf{z}) + \eta \log q_{\boldsymbol{\theta}}(\mathbf{z}|\mathbf{x})}{\eta + \beta}\right)\Big]d\mathbf{x} + \epsilon_\eta \quad (18)$$

$$= \int_{\mathbf{x}} p(\mathbf{x})\Big[\sum_{\mathbf{z}} \log \frac{q_{\boldsymbol{\theta}}(\mathbf{z}|\mathbf{x})}{\tilde{q}^*(\mathbf{z}|\mathbf{x})} q^*(\mathbf{z}|\mathbf{x}) + \log Z_{q^*}\Big]d\mathbf{x} + \epsilon_\eta$$

$$= \epsilon_\eta - \mathbb{E}_{\mathbf{x} \sim p(\mathbf{x})}\Big[D_{\text{KL}}\Big(q^*(\mathbf{z}|\mathbf{x}) \,\|\, q_{\boldsymbol{\theta}}(\mathbf{z}|\mathbf{x})\Big)\Big]$$

It can be seen that computing this value requires computing the partition function, $Z_{q^*}$, and this requires enumerating over all possible sequences of $\mathbf{z}$. Similarly, the outer summation cannot be easily computed, rendering the optimization of $\eta$ using the KL divergence difficult in our setting. For this reason, we make use of the Effective Sample Size to optimize $\eta$, as described below.

## A.2 ESS-BASED TRUST REGION

**Setup.** Given an arbitrary datapoint $\mathbf{x}$, let $q_\theta(\mathbf{z}|\mathbf{x})$ be the proposal distribution. Let $\tilde{q}^*(\mathbf{z}|\mathbf{x})$ denote the *unnormalized* target from Eq. 5, with partition function

$$Z(\mathbf{x}) = \sum_{\mathbf{z}} \tilde{q}^*(\mathbf{z}|\mathbf{x}), \qquad q^*(\mathbf{z}|\mathbf{x}) = \frac{\tilde{q}^*(\mathbf{z}|\mathbf{x})}{Z(\mathbf{x})}.$$

Drawing $\mathbf{z}_{1:K} \sim q_\theta(\cdot|\mathbf{x})$, define the normalized importance weights

$$w_k = \frac{q^*(\mathbf{z}_k|\mathbf{x})}{q_\theta(\mathbf{z}_k|\mathbf{x})}.$$

**Definition.** The effective sample size (ESS) associated with weights $\{w_k\}_{k=1}^K$ is

$$\widehat{\mathrm{ESS}}_K(\mathbf{x}) \;=\; \frac{\left(\sum_{k=1}^K w_k\right)^2}{\sum_{k=1}^K w_k^2}.$$

**Scale invariance.** If instead we had used the unnormalized target $\tilde{q}^*$, then

$$\tilde{w}_k \;=\; \frac{\tilde{q}^*(\mathbf{z}_k|\mathbf{x})}{q_\theta(\mathbf{z}_k|\mathbf{x})} = Z(\mathbf{x})\, w_k.$$

The ESS is invariant to such rescaling:

$$\frac{(\sum_k \tilde{w}_k)^2}{\sum_k \tilde{w}_k^2} = \frac{(Z\sum_k w_k)^2}{Z^2 \sum_k w_k^2} = \frac{(\sum_k w_k)^2}{\sum_k w_k^2}.$$

Thus one may compute $\widehat{\mathrm{ESS}}_K$ using either $q^*$ or $\tilde{q}^*$.

**Normalized ESS ratio.** For convenience, we define the *normalized ESS ratio*

$$\widehat{\rho}_K(\mathbf{x}) \;=\; \frac{1}{K}\,\widehat{\mathrm{ESS}}_K(\mathbf{x}) \;=\; \frac{1}{K}\,\frac{(\sum_{k=1}^K w_k)^2}{\sum_{k=1}^K w_k^2} \;=\; \frac{\left(\frac{1}{K}\sum_{k=1}^K w_k\right)^2}{\frac{1}{K}\sum_{k=1}^K w_k^2}.$$

This form makes both numerator and denominator empirical means, so the strong law of large numbers applies directly.

**Definition (Rényi–2 divergence).** For two distributions $p, q$ with $p$ absolutely continuous w.r.t. $q$, the order–2 Rényi divergence is

$$D_2(p\|q) \;=\; \log \sum_{\mathbf{z}} \frac{p(\mathbf{z})^2}{q(\mathbf{z})}.$$

**Lemma 1** (Population ESS and Rényi–2). *Assume $\mathbb{E}_{q_\theta}[w^2] < \infty$, which holds in the case of finite discrete distributions with full support. Then, as $K \to \infty$,*

$$\widehat{\rho}_K(\mathbf{x}) \;\xrightarrow{a.s.}\; \rho(\mathbf{x}) = \frac{(\mathbb{E}_{q_\theta}[w])^2}{\mathbb{E}_{q_\theta}[w^2]} = \frac{1}{\mathbb{E}_{q_\theta}[w^2]} = \exp\big(-D_2(q^*\|q_\theta)\big).$$

*Proof.* By the strong law,

$$\frac{1}{K}\sum_{k=1}^K w_k \;\to\; \mathbb{E}_{q_\theta}[w], \qquad \frac{1}{K}\sum_{k=1}^K w_k^2 \;\to\; \mathbb{E}_{q_\theta}[w^2],$$

almost surely. Since $q^*$ is normalized,

$$\mathbb{E}_{q_\theta}[w] = \sum_{\mathbf{z}} q_\theta(\mathbf{z}|\mathbf{x})\, \frac{q^*(\mathbf{z}|\mathbf{x})}{q_\theta(\mathbf{z}|\mathbf{x})} = 1.$$

Moreover, by the definition of $D_2$,

$$\mathbb{E}_{q_\theta}[w^2] = \sum_{\mathbf{z}} q_\theta(\mathbf{z}|\mathbf{x}) \Big(\frac{q^*(\mathbf{z}|\mathbf{x})}{q_\theta(\mathbf{z}|\mathbf{x})}\Big)^2 = \sum_{\mathbf{z}} \frac{q^*(\mathbf{z}|\mathbf{x})^2}{q_\theta(\mathbf{z}|\mathbf{x})} = \exp\big(D_2(q^*\|q_\theta)\big).$$

Therefore $\rho(\mathbf{x}) = 1/\mathbb{E}[w^2] = \exp(-D_2(q^*\|q_\theta))$. $\qquad\square$

**Corollary 1** (KL trust region from ESS target). *By monotonicity of Rényi divergences in their order (Van Erven & Harremos, 2014),*

$$D_{\mathrm{KL}}(q^*\|q_\theta) = D_1(q^*\|q_\theta) \;\leq\; D_2(q^*\|q_\theta).$$

*Hence, if $\widehat{\rho}_K(\mathbf{x})$ is adapted to match a target level $\rho_{\mathrm{target}}$, then the update satisfies the KL trust-region bound*

$$D_{\mathrm{KL}}(q^*\|q_\theta) \;\leq\; -\log \rho_{\mathrm{target}}.$$

**Remarks.** *(i) $\widehat{\rho}_K$ is a biased finite-sample estimator of the population $\rho(\mathbf{x})$, but converges almost surely by the strong law. (ii) In practice, we treat $\eta$ as a trainable parameter and update it with stochastic gradient descent on $(\widehat{\rho}_K - \rho_{\mathrm{target}})^2$, thereby driving $\widehat{\rho}_K$ toward $\rho_{\mathrm{target}}$.*

# B  TRAINING DYNAMICS AND STABILITY

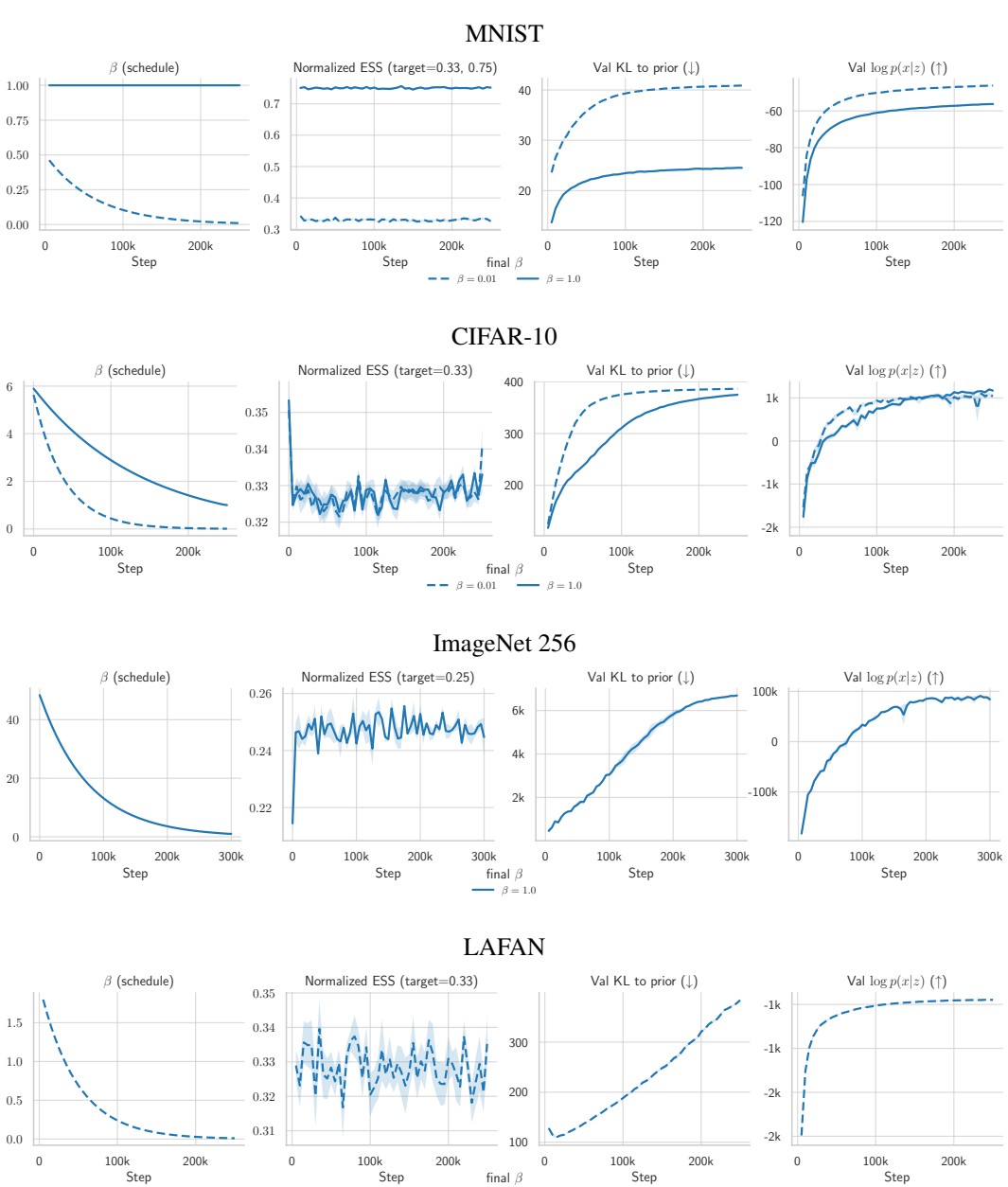

Figure 7: A summary of key metrics for DAPS throughout training.

## C  ABLATIONS: $\beta$ AND ESS-TARGET (CIFAR-10)

We scan $\beta \in \{0.1, 1, 5, 10\}$ and ESS-target $\in \{K/4, K/2, 3K/4\}$ and report the metrics below.

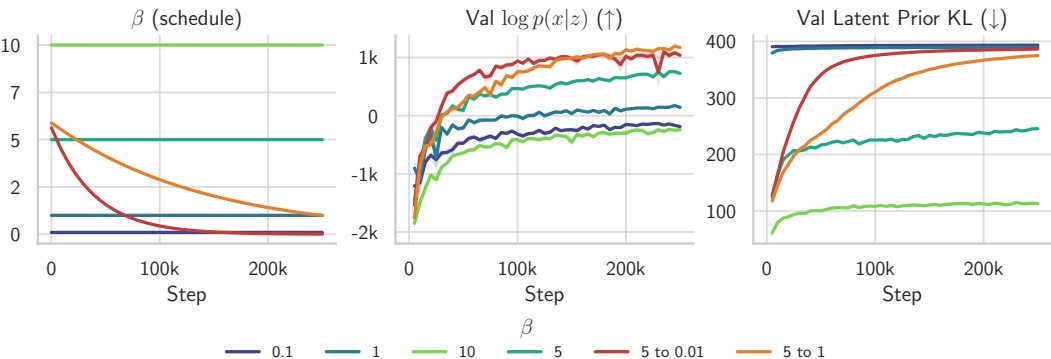

Figure 8: (1) An overview of different $\beta$ values, averaged across the 3 different ESS targets. (2) The scheduled $\beta$ curves are averaged over the 10 seeds from the final experiments, which use an ESS target of 0.33.

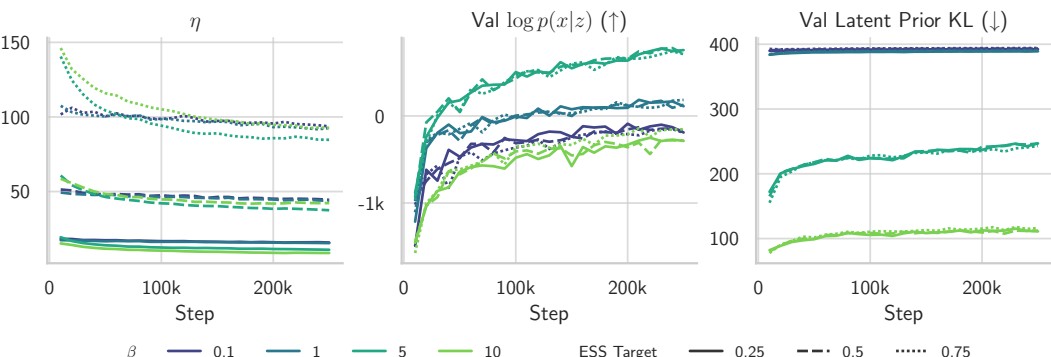

Figure 9: The joint performance of various $\beta$ and ESS combinations. It can be seen that the choice of ESS target has the desired effect, namely, the attenuation of the $\eta$ trajectory throughout training.

## D  ABLATION: NUMBER OF BASELINE SAMPLES (MNIST)

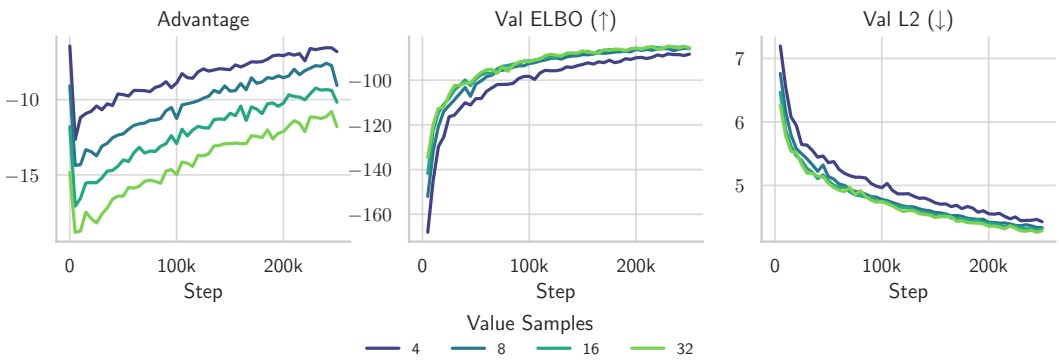

Figure 10: Key metrics w.r.t the number of samples used in the advantage estimation.

# E SUMMARY OF KEY METRICS

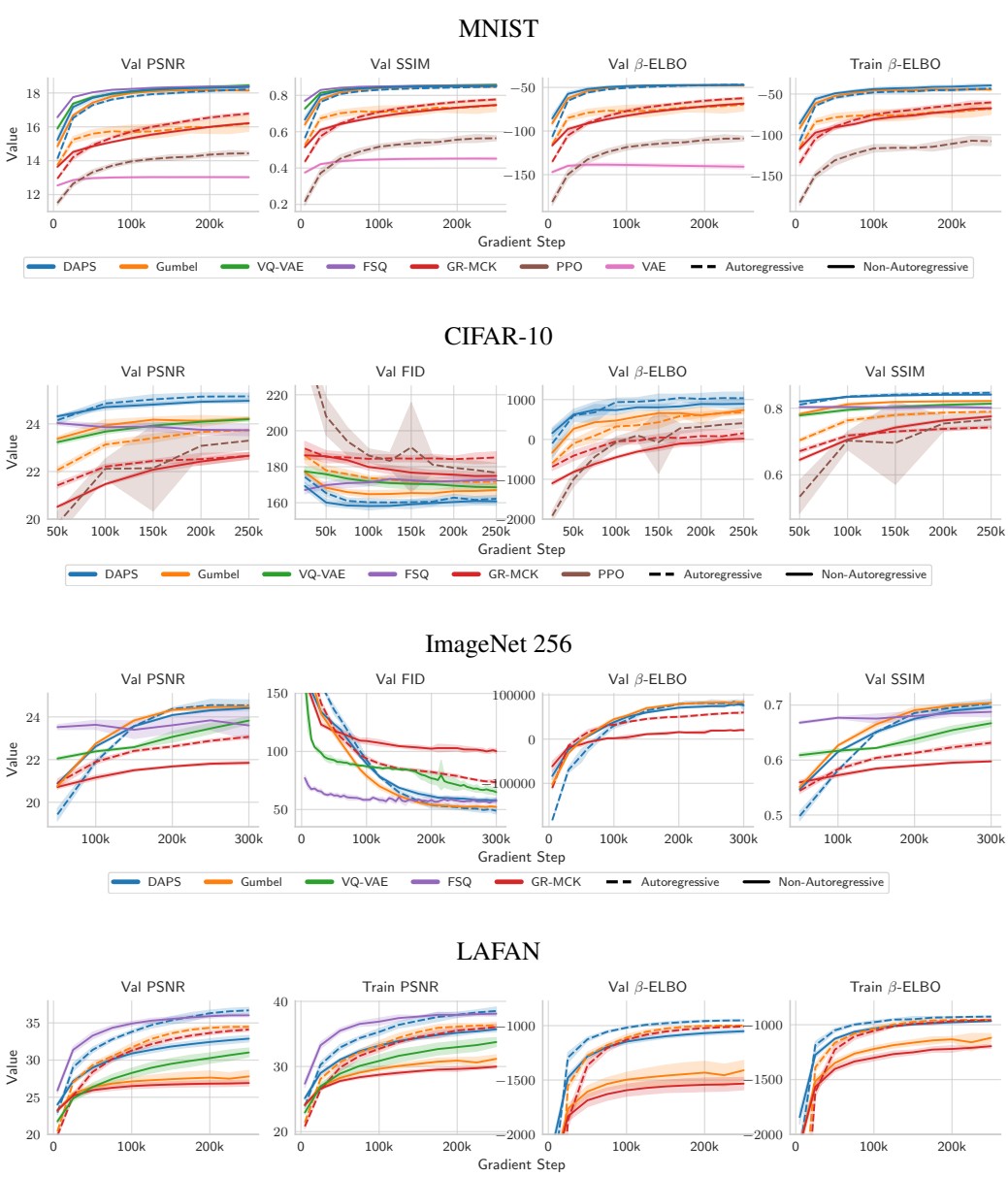

Figure 11: A summary of key metrics averaged over all seeds (for $\beta = 0.01$), $\pm$ 1 SD.

## F  BOTTLENECK SCALING EXPERIMENTS

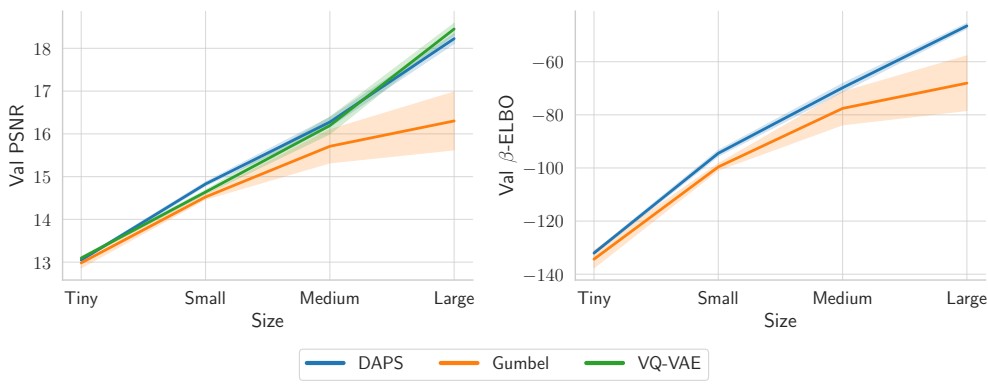

Figure 12: Across bottleneck sizes (tiny: $B=4$, $V=4$ = 8 bits; small: $B=4$, $V=16$ = 16 bits; medium: $B=8$, $V=16$ = 32 bits; large: $B=8$, $V=256$ = 64 bits), DAPS exhibits consistent scaling behavior on MNIST. In the left plot, validation PSNR increases predictably with latent capacity (bits = $B \log_2 V$), and DAPS remains competitive with VQ-VAE across all settings. Notably, VQ-VAE does not incorporate an entropy term, whereas DAPS (trained with $\beta = 0.01$ for all sizes) includes mild entropy regularization, so the comparison reflects slightly different optimization objectives. In the right plot, DAPS generally outperforms Gumbel across all bottleneck sizes in validation $\beta$-ELBO, with the gap widening as capacity increases. Together, these results verify that DAPS scales as expected with latent capacity while maintaining sufficient reconstruction performance.

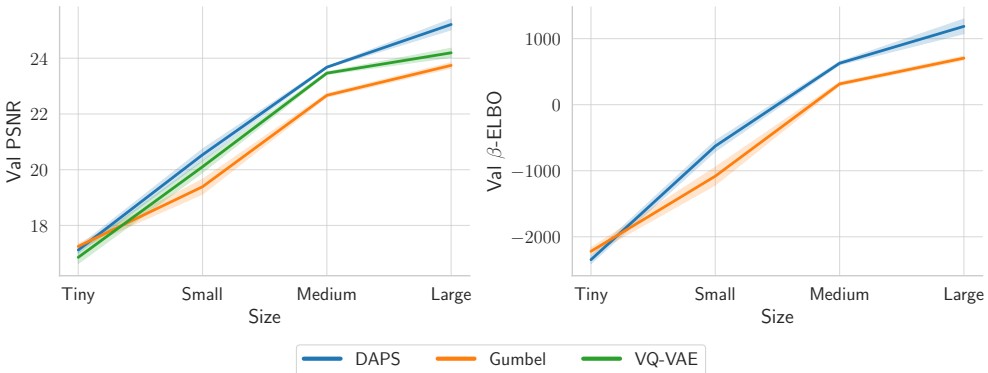

Figure 13: Across bottleneck sizes (tiny: $B=4$, $V=512$ = 36 bits; small: $B=16$, $V=512$ = 144 bits; medium: $B=64$, $V=64$ = 384 bits; large: $B=64$, $V=512$ = 576 bits), DAPS exhibits strong scaling behavior on CIFAR. In the left plot, validation PSNR improves with latent capacity, and DAPS remains competitive with, and slightly outperforms, VQ-VAE as the bottleneck increases. A similar trend to Fig. 12 appears in the right plot: DAPS generally outperforms Gumbel in validation $\beta$-ELBO, while not exhibiting an advantage at the tiny bottleneck size. For all models and bottleneck configurations in Figs. 12 and 13, we performed a hyperparameter sweep and report the best-performing setting for each. We use 5 seeds for the Tiny, Small, and Medium runs, and the 10 seeds from our main experiments for the Large runs.

# G  LATENT CODE UTILIZATION

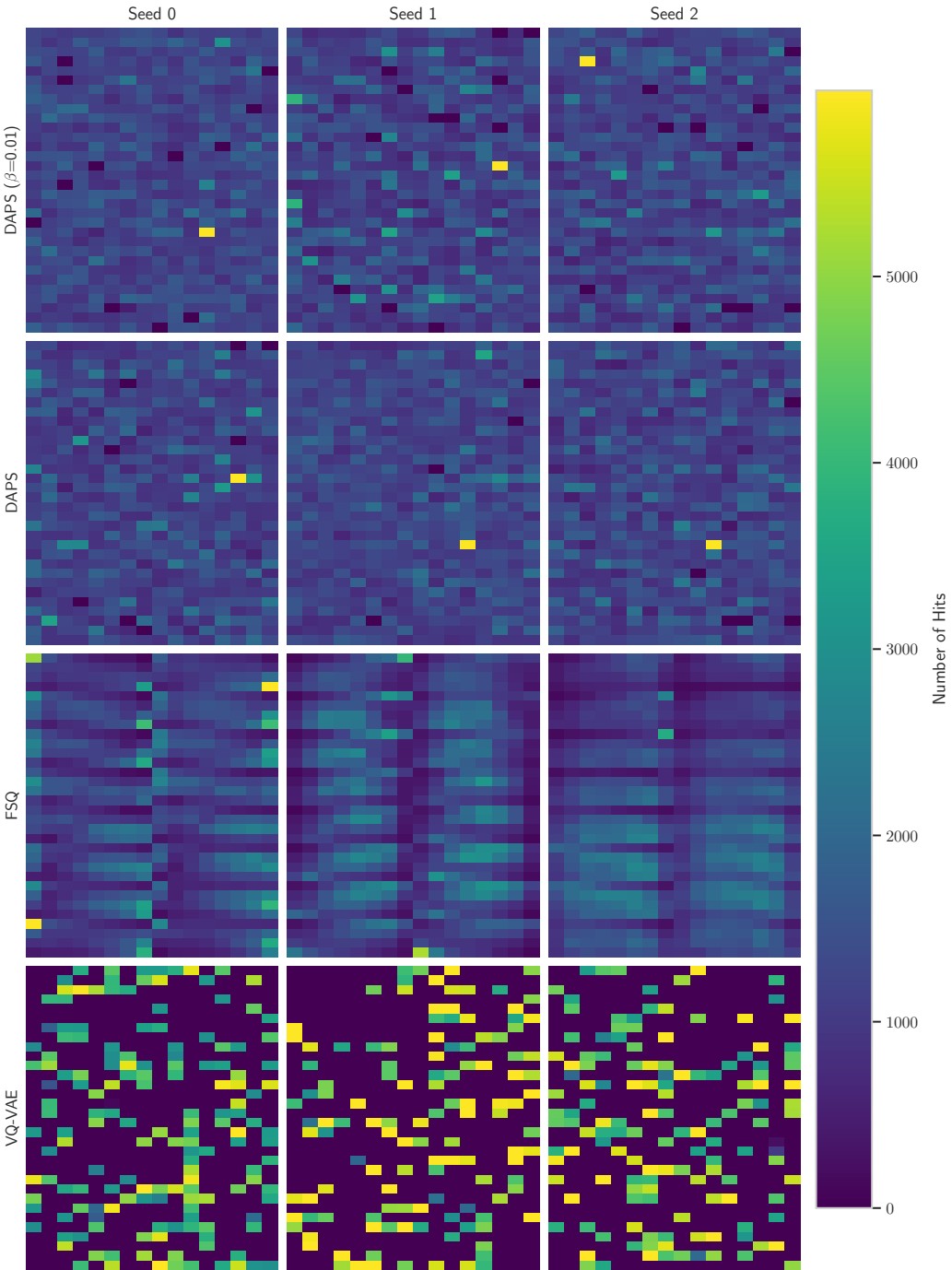

Figure 14: Marginal (empirical) distribution of $q(\mathbf{z}_i)$ on the full CIFAR validation set. Each grid shows the full vocabulary $V$ of the recognition model, reshaped for visualization, with cell values indicating the number of times each latent code is used. DAPS effectively utilizes all codes, whereas VQ-VAE struggles to do so. Although FSQ aims to address this limitation and indeed achieves stronger utilization, its vocabulary usage remains much sparser than DAPS, highlighting the favorable entropy properties induced by our objective.

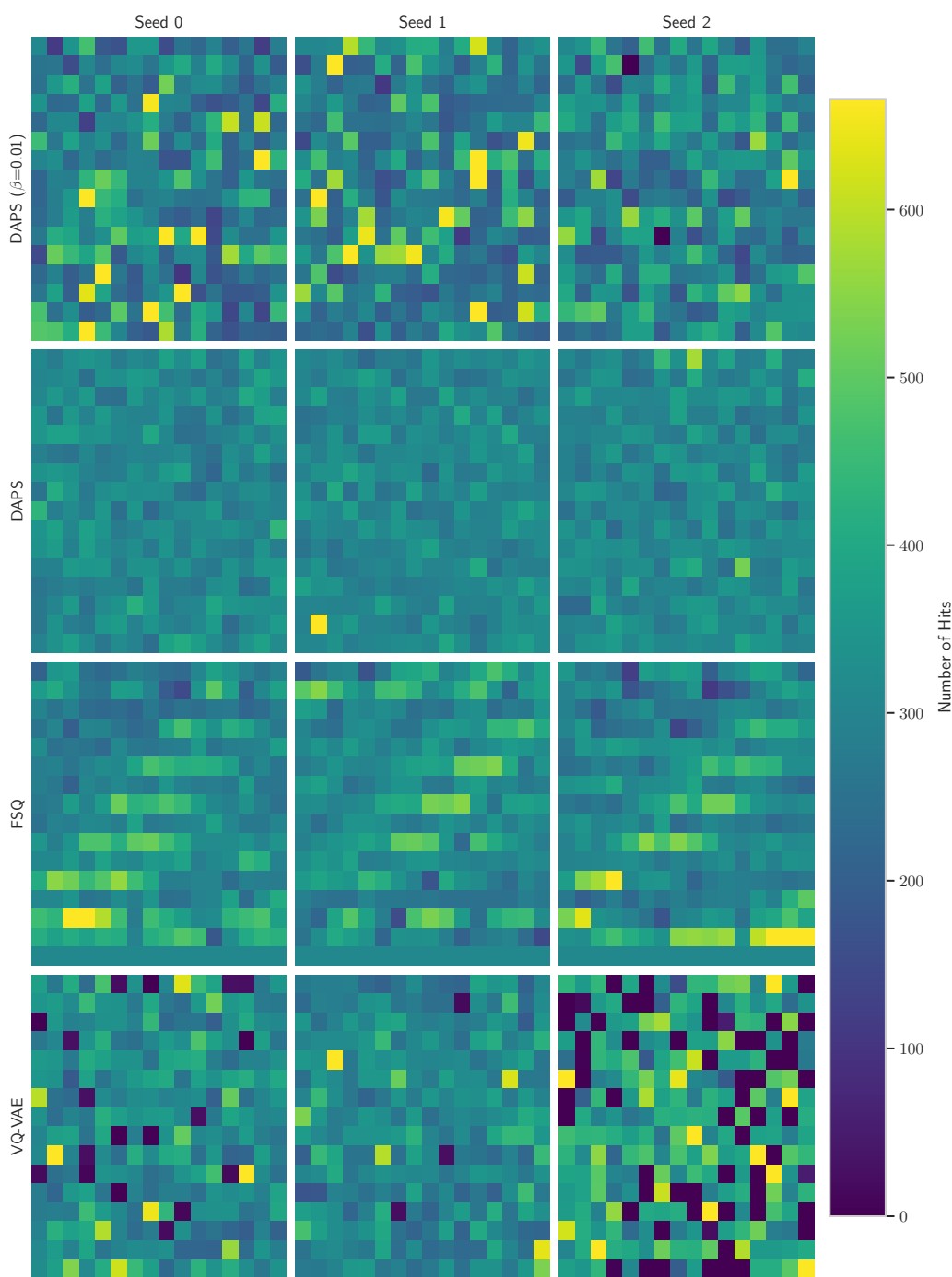

Figure 15: Marginal (empirical) distribution of $q(\mathbf{z}_i)$ on the full MNIST validation set. Each grid shows the full vocabulary $V$ of the recognition model, reshaped for visualization, with cell values indicating latent-code usage. DAPS again achieves broad and balanced utilization of the vocabulary, whereas FSQ exhibits noticeably sparser usage and VQ-VAE contains several low-hit regions. This further highlights the favorable entropy properties promoted by our objective.

# H COMPUTATIONAL OVERVIEW

We provide a brief comparison of the computational characteristics of the methods considered, including memory usage, parameter counts (for both recognition and generative models), and training speed (measured in gradient steps per minute).

## H.1 MODEL PARAMETER SUMMARY

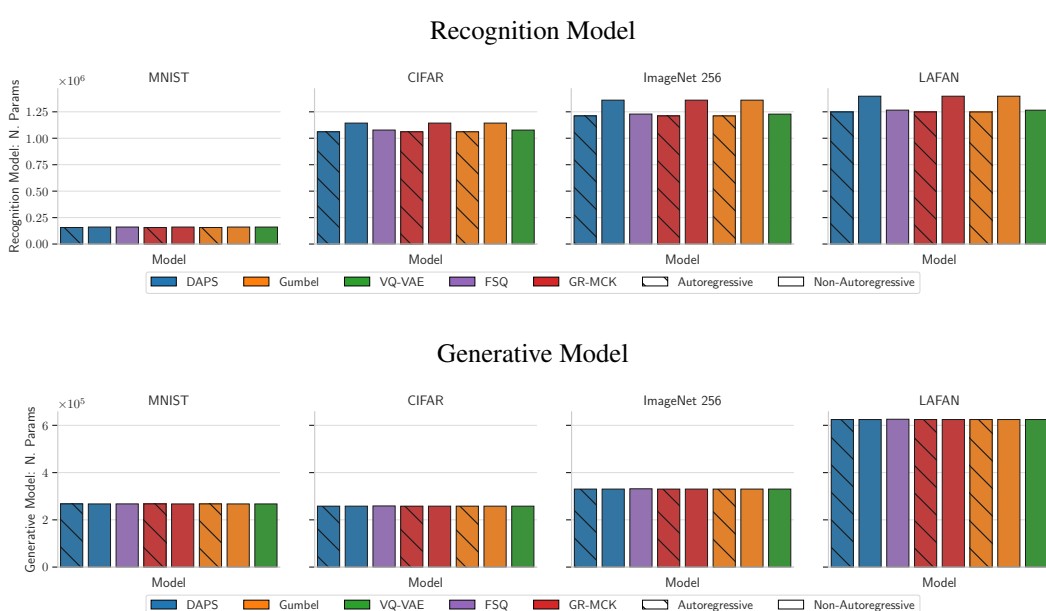

Figure 16: A comparison of model parameter counts.

## H.2 AUTOREGRESSIVE MODELS: RESOURCE USAGE AND SPEED

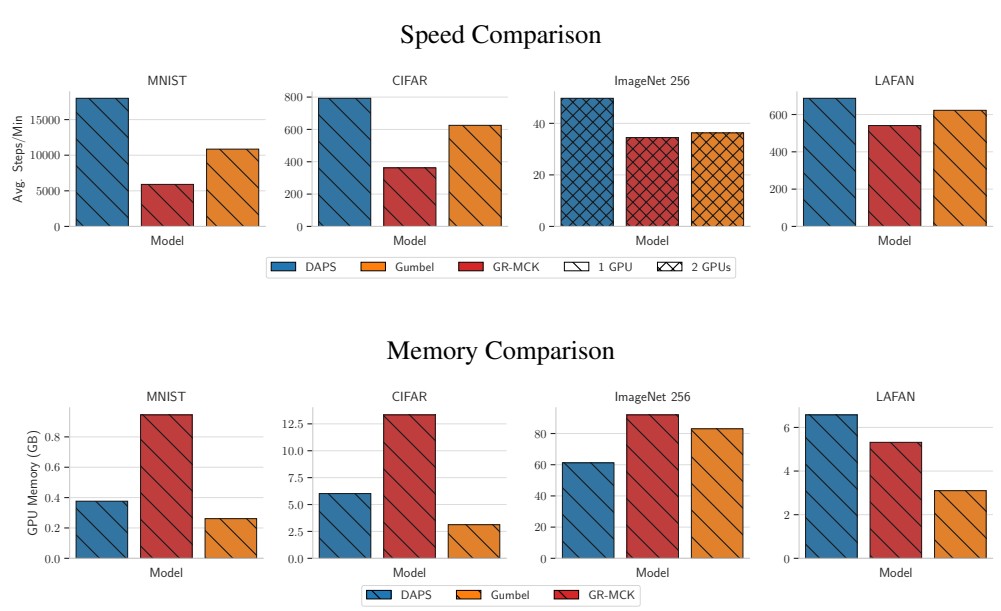

Figure 17: A comparison of resource usage and speed for autoregressive models.

## H.3 Non-Autoregressive Models: Resource Usage and Speed

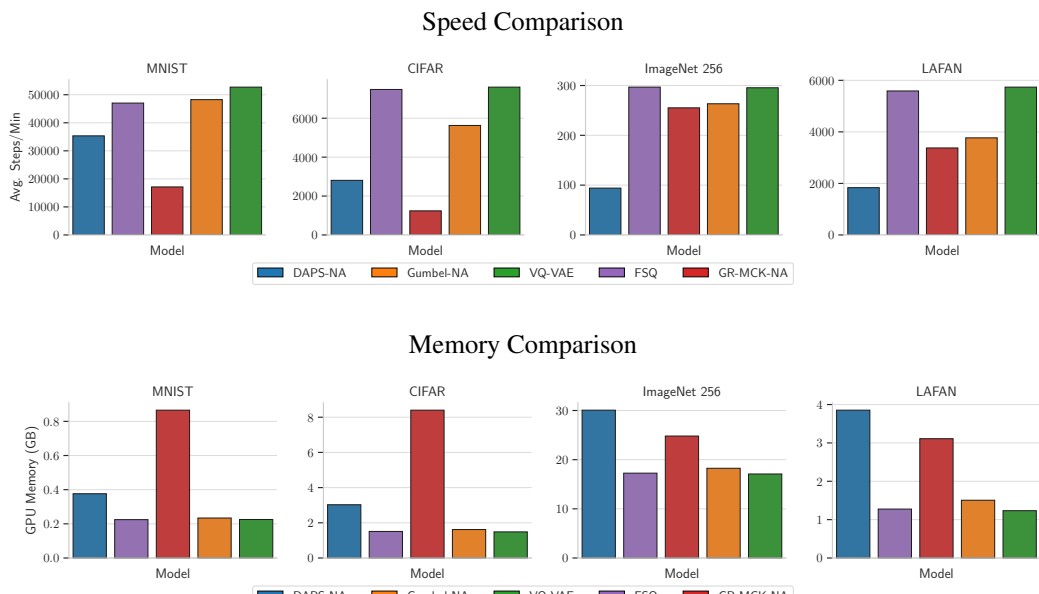

Figure 18: A comparison of resource usage and speed for non-autoregressive models.

## H.4 Computational Considerations

**Model parameter comparison (Figure 16).** We compare the parameter counts of the recognition and generative models across methods. All baselines use parameter budgets that are comparable to, or larger than, those of our primary model (DAPS–autoregressive), ensuring a fair evaluation. Minor deviations reflect architectural differences inherent to each method.

**Autoregressive resource usage (Figure 17).** We next compare memory usage and training speed for the autoregressive variants. On ImageNet 256, both Gumbel and GR-MCK exceed the memory capacity of a single GPU and must be trained across two GPUs; in contrast, DAPS fits the full batch on a single A100, though we use 2 GPUs in practice. Although we do not evaluate Gumbel on ImageNet due to instability, we report its memory and speed characteristics for completeness. Memory usage for GR-MCK is dominated by the MC-$K$ factor, explaining its substantially larger footprint relative to Gumbel. We note that only 8 value samples are used for LAFAN in Fig. 17 and 18, which slightly degrades performance (PSNR=35.19 and $\beta$-ELBO=-985 for DAPS; PSNR=32.50, $\beta$-elbo=-1067 for DAPS-NA). Overall, we observe that DAPS scales more favorably on ImageNet compared to Gumbel, which we hypothesize is related to the interplay between backpropagation through time (BPTT) gradients and the quadratic scaling properties of transformers. These results highlight that implementing DAPS is practical even at larger bottleneck sizes.

**Non-autoregressive resource usage (Figure 18).** As expected, we observe that DAPS-NA is faster and more memory-efficient than its autoregressive counterpart. However, DAPS-NA is generally slower than baseline methods. Nonetheless, the computational cost remains practical in absolute terms (e.g., under 10 minutes on MNIST and under 2 hours on CIFAR for 250k gradient steps). However, we focus on the autoregressive variant throughout the paper, given that DAPS-NA generally underperforms autoregressive DAPS. Finally, we again observe that the MC-$K$ parameter drives GR-MCK's memory and speed profile. Overall, both variants of DAPS achieve better performance at the cost of higher computation. However, this does not prevent scaling to challenging data sets such as ImageNet, even for the autoregressive variant.

# I  VISION MODEL ARCHITECTURE AND PARAMETER SUMMARY

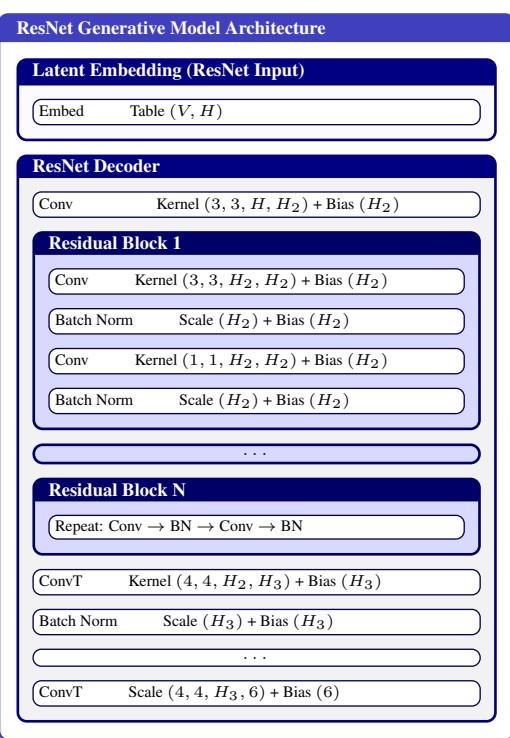

**Vision Recognition Model Architecture**

**Patch Feed Forward (Encoder Input)**

Dense        Kernel $(D_{\text{patch}}, H)$ + Bias $(H)$

**Latent Embedding (Decoder Input)**

Embed        Table $(V, H)$

**Transformer**

**Encoder Block 1**

Layer Norm        Scale $(H)$ + Bias $(H)$

**MultiHead Attention**

Query $(H, N_h, H/N_h)$ | Key $(H, N_h, H/N_h)$
Value $(H, N_h, H/N_h)$ | Out $(N_h, H/N_h, H)$

Layer Norm        Scale $(H)$ + Bias $(H)$

Dense        Kernel $(H, HR)$ + Bias $(HR)$

Dense        Kernel $(HR, H)$ + Bias $(H)$

$\ldots$

**Encoder Block N**

Repeat: LN → MHA → LN → MLP

**Encoder–Decoder Block 1**

Layer Norm        Scale $(H)$ + Bias $(H)$

**MultiHead Attention 1**

Query $(H, N_h, H/N_h)$ | Key $(H, N_h, H/N_h)$
Value $(H, N_h, H/N_h)$ | Out $(N_h, H/N_h, H)$

Layer Norm        Scale $(H)$ + Bias $(H)$

**MultiHead Attention 2**

Query $(H, N_h, H/N_h)$ | Key $(H, N_h, H/N_h)$
Value $(H, N_h, H/N_h)$ | Out $(N_h, H/N_h, H)$

Layer Norm        Scale $(H)$ + Bias $(H)$

Dense        Kernel $(H, HR)$ + Bias $(HR)$

Dense        Kernel $(HR, H)$ + Bias $(H)$

$\ldots$

**Encoder–Decoder Block N**

Repeat: LN → MHA → LN → X-MHA → LN → MLP

Layer Norm        Scale $(H)$ + Bias $(H)$

Dense        Kernel $(H, V)$ + Bias $(V)$

**ResNet Generative Model Architecture**

**Latent Embedding (ResNet Input)**

Embed        Table $(V, H)$

**ResNet Decoder**

Conv        Kernel $(3, 3, H, H_2)$ + Bias $(H_2)$

**Residual Block 1**

Conv        Kernel $(3, 3, H_2, H_2)$ + Bias $(H_2)$

Batch Norm        Scale $(H_2)$ + Bias $(H_2)$

Conv        Kernel $(1, 1, H_2, H_2)$ + Bias $(H_2)$

Batch Norm        Scale $(H_2)$ + Bias $(H_2)$

$\ldots$

**Residual Block N**

Repeat: Conv → BN → Conv → BN

ConvT        Kernel $(4, 4, H_2, H_3)$ + Bias $(H_3)$

Batch Norm        Scale $(H_3)$ + Bias $(H_3)$

$\ldots$

ConvT        Scale $(4, 4, H_3, 6)$ + Bias $(6)$

**Generative Model.** The vision generative model architecture is a standard ResNet decoder. This model takes as input a latent sequence (of length 64 for CIFAR-10, and length $1,024$ for ImageNet) and converts it to a sequence of embeddings using the embedding table. This is then reshaped into a square tensor of embeddings ($8 \times 8 \times 128$ for CIFAR-10, and $32 \times 32 \times 128$ for ImageNet) to be processed by the ResNet. After which, convolutional transpose layers are used to upsample the tensor to the final image dimension, including an additional 3 channels for the RGB variance. See I.1 and I.2 for the corresponding variable values.

**Recognition Model.** The recognition model architecture is a standard vision transformer. Cross attention is used for autoregressive-based methods, while self-attention is used for VQ-based methods. We also tried using the ResNet encoder proposed in VQ-VAE, but found no performance benefit. See I.1 and I.2 for the corresponding variable values.

## I.1 IMAGENET-256 VISION HYPERPARAMETERS

```
Config (ImageNet-256)

model:
  block_size: 1024                   # B
  vocab_size: 1024                   # V
  embed_dim: 128                     # H

recognition_model:
  transformer:
    num_heads: 4                     # N_h
    num_layers:                      # N
      autoregressive: 2
      non_autoregressive: 6
    mlp_ratio: 4                     # R
    qkv_dim: 128
  num_patches: 1024                  # 32x32 patches
  patch_ffwd: [dense(128)]

generative_model:
  resnet:
    hidden_dim: 64                   # H_2
    activation: relu
    norm: batch_norm
    residual:
      num_blocks: 2                  # N
      hidden_dim: 64                 # H_2
      activation: relu
      norm: batch_norm
```

**Network Hyperparameter Summary.** The configuration above lists the architecture-specific hyperparameters used for training ImageNet 256 (shared by all methods). Variable names, indicated by comments on the right-hand side, correspond to the variables in the Vision Transformer Diagram featured in I. The autoregressive sampling used by ELBO-based methods can be sped up through the use of KV caching, as past activations do not need to be recalculated on subsequent forward passes. Our implementation uses a decoder cache window size of 8 blocks.

| Method (final $\beta$) | Commit. coeff. | Temp. | Init. $\beta$ | Target ESS$_\rho$ | MC-$K$ | $K$ (DAPS) | LR |
|---|---|---|---|---|---|---|---|
| DAPS   (1.0) | – | – | 50 | 0.25 | – | 8 | 3e−4 |
| DAPS-NA   (1.0) | – | – | 50 | 0.25 | – | – | 3e−4 |
| GRMCK   (1.0) | – | $1.0 \to 0.75$ | 10 | – | 10 | – | 1e−4 |
| GRMCK-NA   (1.0) | – | $1.0 \to 0.1$ | 10 | – | 10 | – | 3e−4 |
| Gumbel-NA   (1.0) | – | $1.0 \to 0.1$ | 50 | – | – | – | 3e−4 |
| FSQ | 0.25 | – | – | – | – | – | 3e−4 |
| VQVAE | 0.25 | – | – | – | – | – | 3e−4 |

Table 3: ImageNet256 model-specific hyperparameters (final $\beta = 1.0$).

## I.2 CIFAR-10 VISION HYPERPARAMETERS

```
Config (CIFAR-10)

model:
  block_size: 64                    # B
  vocab_size: 512                   # V
  embed_dim: 128                    # H

recognition_model:
  transformer:
    num_heads: 4                    # N_h
    num_layers:                     # N
      autoregressive: 2
      non_autoregressive: 4
    mlp_ratio:                      # R
      autoregressive: 4
      non_autoregressive: 6
    qkv_dim: 128
  num_image_patches: 64             # 8x8 patches
  patch_ffwd: [dense(128)]

generative_model:
  resnet:
    hidden_dim: 64                  # H_2
    activation: relu
    norm: batch_norm
    residual:
      num_blocks: 2                 # N
      hidden_dim: 64                # H_2
      activation: relu
      norm: batch_norm
```

**Network Hyperparameter Summary.** The configuration above lists the architecture-specific hyperparameters used for training CIFAR-10 (shared by all methods). Variable names, indicated by comments on the right-hand side, correspond to the variables in the Vision Transformer Diagram featured in I. This design follows the architectures presented in Van Den Oord et al. (2017) and Alexey (2020).

| Method (final $\beta$) | Commit. coeff. | Temp. | Init. $\beta$ | Target $\mathrm{ESS}_\rho$ | MC-$K$ | $K$ (DAPS) | LR |
|---|---|---|---|---|---|---|---|
| DAPS  (0.01) | – | – | 6 | 0.33 | – | 8 | 3e−4 |
| DAPS-NA  (0.01) | – | – | 6 | 0.33 | – | 8 | 3e−4 |
| GRMCK  (0.01) | – | $1.0 \rightarrow 0.75$ | 3 | – | 100 | – | 3e−4 |
| GRMCK-NA  (0.01) | – | $1.0 \rightarrow 0.1$ | 3 | – | 100 | – | 3e−4 |
| Gumbel  (0.01) | – | 1.0 | 6 | – | – | – | 1e−4 $\rightarrow$ 1e−5 |
| Gumbel-NA  (0.01) | – | $1.0 \rightarrow 0.1$ | 6 | – | – | – | 3e−4 |
| FSQ | 0.25 | – | – | – | – | – | 3e−4 |
| VQVAE | 0.25 | – | – | – | – | – | 3e−4 |
| DAPS  (1.0) | – | – | 6 | 0.33 | – | 8 | 3e−4 |
| DAPS-NA  (1.0) | – | – | 6 | 0.33 | – | 8 | 3e−4 |
| GRMCK  (1.0) | – | $1.0 \rightarrow 0.75$ | 3 | – | 100 | – | 3e−4 |
| GRMCK-NA  (1.0) | – | $1.0 \rightarrow 0.75$ | 3 | – | 100 | – | 3e−4 |
| Gumbel  (1.0) | – | $1.0 \rightarrow 0.75$ | 6 | – | – | – | 3e−4 $\rightarrow$ 3e−5 |
| Gumbel-NA  (1.0) | – | $1.0 \rightarrow 0.1$ | 6 | – | – | – | 3e−4 |

Table 4: CIFAR-10 model-specific hyperparameters for both $\beta = 0.01$ and $\beta = 1.0$.

## I.3   MNIST Vision Hyperparameters

```
Config (MNIST)

model:
  block_size: 8                          # B
  vocab_size: 256                        # V
  embed_dim: 64                          # H

recognition_model:
  transformer:
    num_heads: 4                         # N_h
    num_layers:                          # N
      autoregressive: 1
      non_autoregressive: 2
    mlp_ratio:                           # R
      autoregressive: 4
      non_autoregressive: 6
      vq_fsq: 7
    qkv_dim: 64
  num_image_patches: 8
  patch_ffwd: [dense(64)]

generative_model:
  ffwd: [dense(64), dense(256)]
  activation: relu
```

**Network Hyperparameter Summary.** The configuration above lists the architecture-specific hyperparameters used for training MNIST. Variable names for the recognition model, indicated by comments on the right-hand side, correspond to variables in the Vision Transformer Diagram featured in I. The generative model is a fully connected, feed-forward network with two hidden layers.

| Method (final $\beta$) | Commit. coeff. | Temp. | Init. $\beta$ | Target ESS$_\rho$ | MC-$K$ | $K$ (DAPS) | LR |
|---|---|---|---|---|---|---|---|
| DAPS   (0.01) | – | – | 0.5 | 0.33 | – | 8 | 3e−4 |
| DAPS-NA   (0.01) | – | – | 0.5 | 0.5 | – | 8 | 3e−4 |
| GRMCK   (0.01) | – | $1.0 \rightarrow 0.75$ | 0.5 | – | 100 | – | 3e−4 |
| GRMCK-NA   (0.01) | – | $1.0 \rightarrow 0.1$ | 0.5 | – | 100 | – | 3e−4 |
| Gumbel   (0.01) | – | 1.0 | 0.5 | – | – | – | 3e−4 |
| Gumbel-NA   (0.01) | – | $1.0 \rightarrow 0.1$ | 0.5 | – | – | – | 3e−4 |
| FSQ | 0.25 | – | – | – | – | – | 3e−4 |
| VQVAE | 0.25 | – | – | – | – | – | 3e−4 |
| DAPS   (1.0) | – | – | 1 | 0.75 | – | 8 | 3e−4 |
| DAPS-NA   (1.0) | – | – | 1 | 0.5 | – | 8 | 3e−4 |
| GRMCK   (1.0) | – | $1.0 \rightarrow 0.75$ | 1 | – | 100 | – | 3e−4 |
| GRMCK-NA   (1.0) | – | $1.0 \rightarrow 0.75$ | 1 | – | 100 | – | 3e−4 |
| Gumbel   (1.0) | – | $1.0 \rightarrow 0.75$ | 1 | – | – | – | 3e−4 |
| Gumbel-NA   (1.0) | – | $1.0 \rightarrow 0.1$ | 1 | – | – | – | 3e−4 |

Table 5: MNIST model-specific hyperparameters for both $\beta = 0.01$ and $\beta = 1.0$.

## J TRAJECTORY MODEL ARCHITECTURE AND PARAMETER SUMMARY

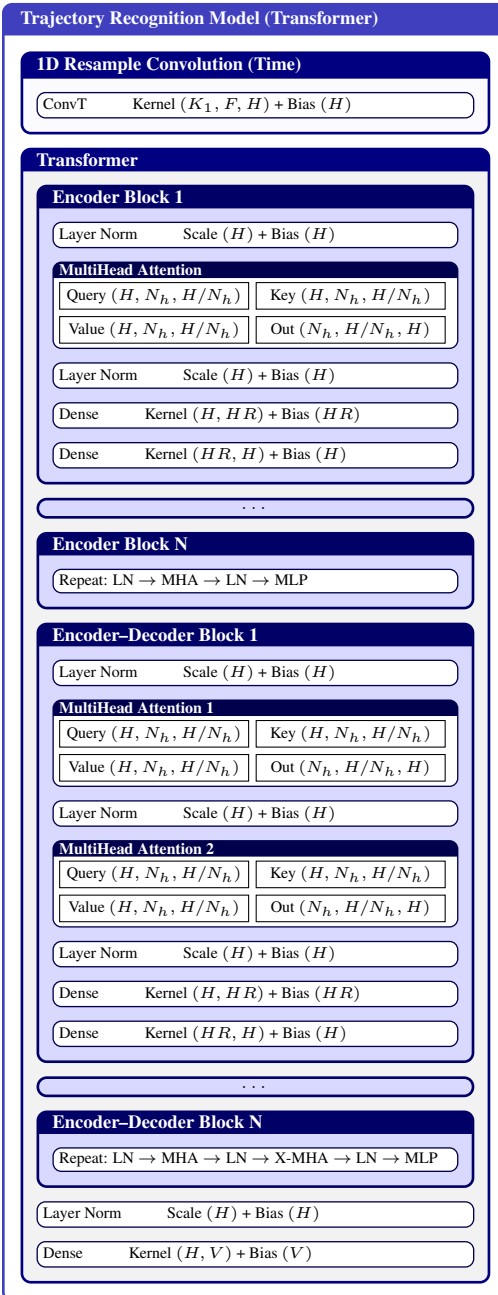

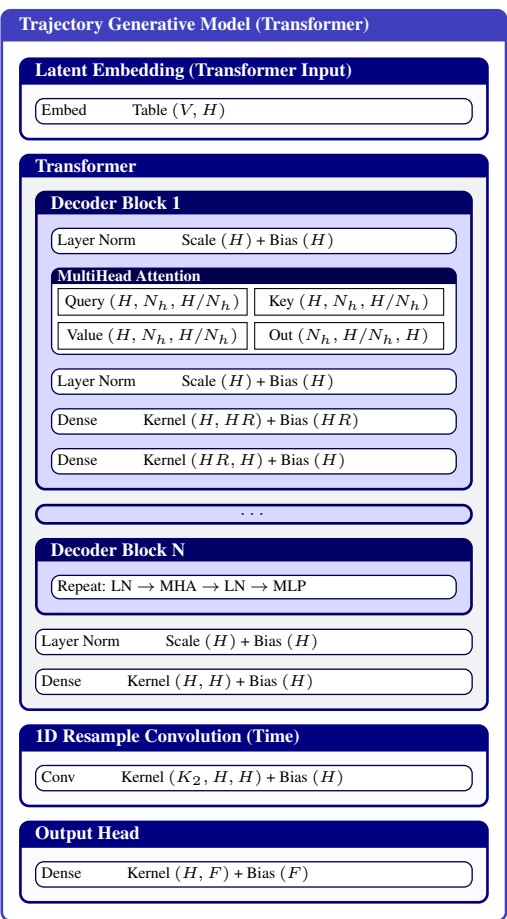

**Generative Model.** The trajectory generative model architecture is similar in symmetry to the trajectory recognition model. After processing the embedded latent sequence, the sequence is resampled by a 1D convolutional network to reach the desired trajectory segment length. The result is a continuous trajectory of length 32 with 127 features per timestep.

**Recognition Model.** The trajectory recognition model architecture shares a similar structure to the vision recognition model. Instead of using an image patcher (as in the ViT), we use a 1D convolutional network to embed the trajectory sequence. Cross attention is used for autoregressive-based methods, while self-attention is used for VQ-based methods.

## J.1 LAFAN TRAJECTORY HYPERPARAMETERS

```
Config (Trajectory: LAFAN)

model:
  block_size: 64                         # B
  vocab_size: 1024                       # V
  embed_dim: 128                         # H

trajectory:
  length: 32                             # Segment length
  feature_dim: 121                       # F

recognition_model:
  resample_conv1d:
    output_length: ${model.block_size}
    output_channels: ${model.embed_dim}
    stride: 2
    kernel_size: 4                       # K_1
  transformer:
    num_heads: 4                         # N_h
    num_layers:                          # N
      autoregressive: 2
      non_autoregressive: 6
    mlp_ratio: 4                         # R
    qkv_dim: 128

generative_model:
  transformer:
    <<: *recognition_model.transformer
  resample_conv1d:
    output_length: ${trajectory.length}
    output_channels: ${model.embed_dim}
    stride: 2
    kernel_size: 4                       # K_2
  output_ffwd: [dense:${trajectory.feature_dim}]
```

**Network Hyperparameter Summary.** The configuration above lists the architecture-specific hyperparameters used for training the LAFAN dataset. Variable names, indicated by comments on the right-hand side, correspond to the variables in the Trajectory Transformer Diagram featured in J. The recognition model and generative models are highly symmetric. These networks reconstruct randomly sampled trajectory segments of length 32 with feature dimension 121. In order to decouple the block size from the length of the trajectory segment, we use a 1D convolutional network to resample the trajectory length to the model's block size. This operation is then reversed at the output of the generative model to achieve the desired segment length.

| Method (final $\beta$) | Commit. coeff. | Temp. | Init. $\beta$ | Target ESS$_\rho$ | MC-$K$ | $K$ (DAPS) | LR |
|---|---|---|---|---|---|---|---|
| DAPS (0.01) | – | – | 2 | 0.33 | – | 16 | 3e−4 |
| DAPS-NA (0.01) | – | – | 1 | 0.33 | – | 16 | 3e−4 |
| GRMCK (0.01) | – | 1.0 | 2 | – | 10 | – | 1e−4 → 1e−5 |
| GRMCK-NA (0.01) | – | 1.0 | 1 | – | 10 | – | 3e−4 |
| Gumbel (0.01) | – | 1.0 | 2 | – | – | – | 1e−4 → 1e−5 |
| Gumbel-NA (0.01) | – | 1.0 → 0.75 | 3 | – | – | – | 3e−4 |
| FSQ | 0.25 | – | – | – | – | – | 3e−4 |
| VQVAE | 0.5 | – | – | – | – | – | 3e−4 |

Table 6: LAFAN model-specific hyperparameters (final $\beta = 0.01$).

## K   DOWNSTREAM ROBOTICS EXPERIMENT DETAILS

**Dancing**

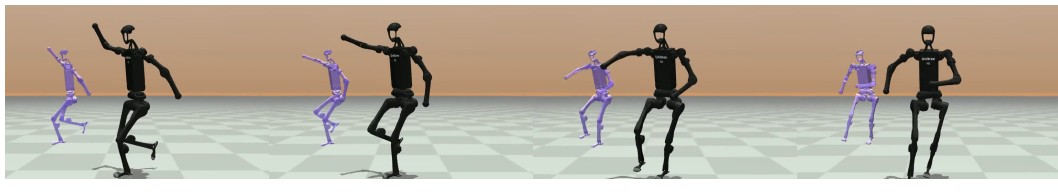

**Running**

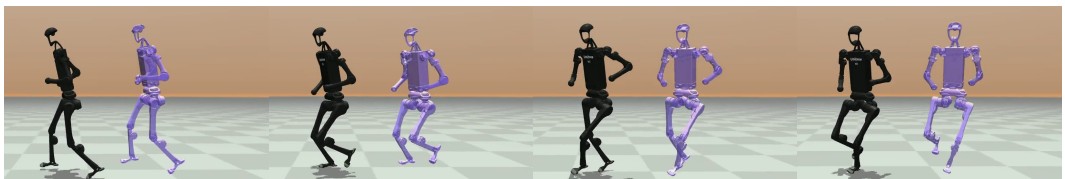

**Walking**

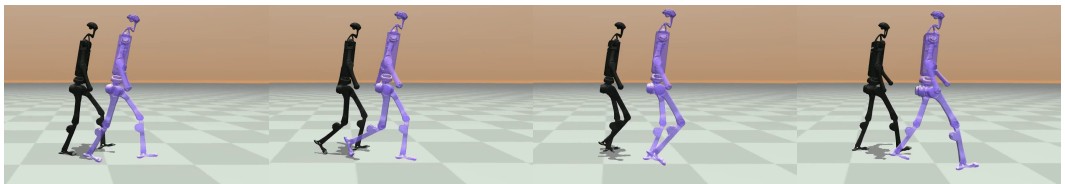

Figure 19: In our downstream experiment, we use different language commands to generate motion styles with DAPS, and use a physical consistent decoder given by an RL policy and a robot simulator to decode the discrete latent representation.

We train a low-level policy with DeepMimic to imitate motions from the LAFAN1 dataset on a Unitree H1 robot simulated in MuJoCo. Unlike the conventional DeepMimic formulation, where motion reference information for the next timestep (site positions, joint angles, and velocities) is explicitly given as a goal to the policy, we provide the policy with a compact embedding of the desired future motion as the goal. We condition the high-level policy—which produces these embeddings—on language information and motion goals. For the former, we use BERT embeddings of a brief description of the motion file. For the latter, we provide the encoder with the last 32 timesteps and future 32 timesteps of the center of mass (COM) trajectory, relative to the robot's base frame. The resulting output is a compact vector of 64 latent indices.

We train the low-level policy conditioned on these embeddings with PPO using the LocoMujoco framework (Al-Hafez et al., 2023). The policy transforms the embedded indices with a 1D convolutional layer, then concatenates the resulting features with standard proprioceptive information: joint angles, joint velocities, IMU information (projected gravity and torso angular velocities), and the previous timestep's action, ultimately feeding these into a multilayer perceptron of size [2048, 1024, 512] with `elu` activations, resulting in a 19-dimensional vector of torques to be applied to each motor of the Unitree H1 robot. The policy is executed at 100 Hz.

## L   DATASETS AND PREPROCESSING

**Images.**   For ImageNet-256, we randomly crop and flip the images. All images are standardized using their empirical mean and standard deviation. CIFAR-10 and MNIST are also standardized similarly.

**Trajectories (LAFAN).**   For the LAFAN dataset, the motion sequences are normalized to have zero mean and unit variance along each feature.

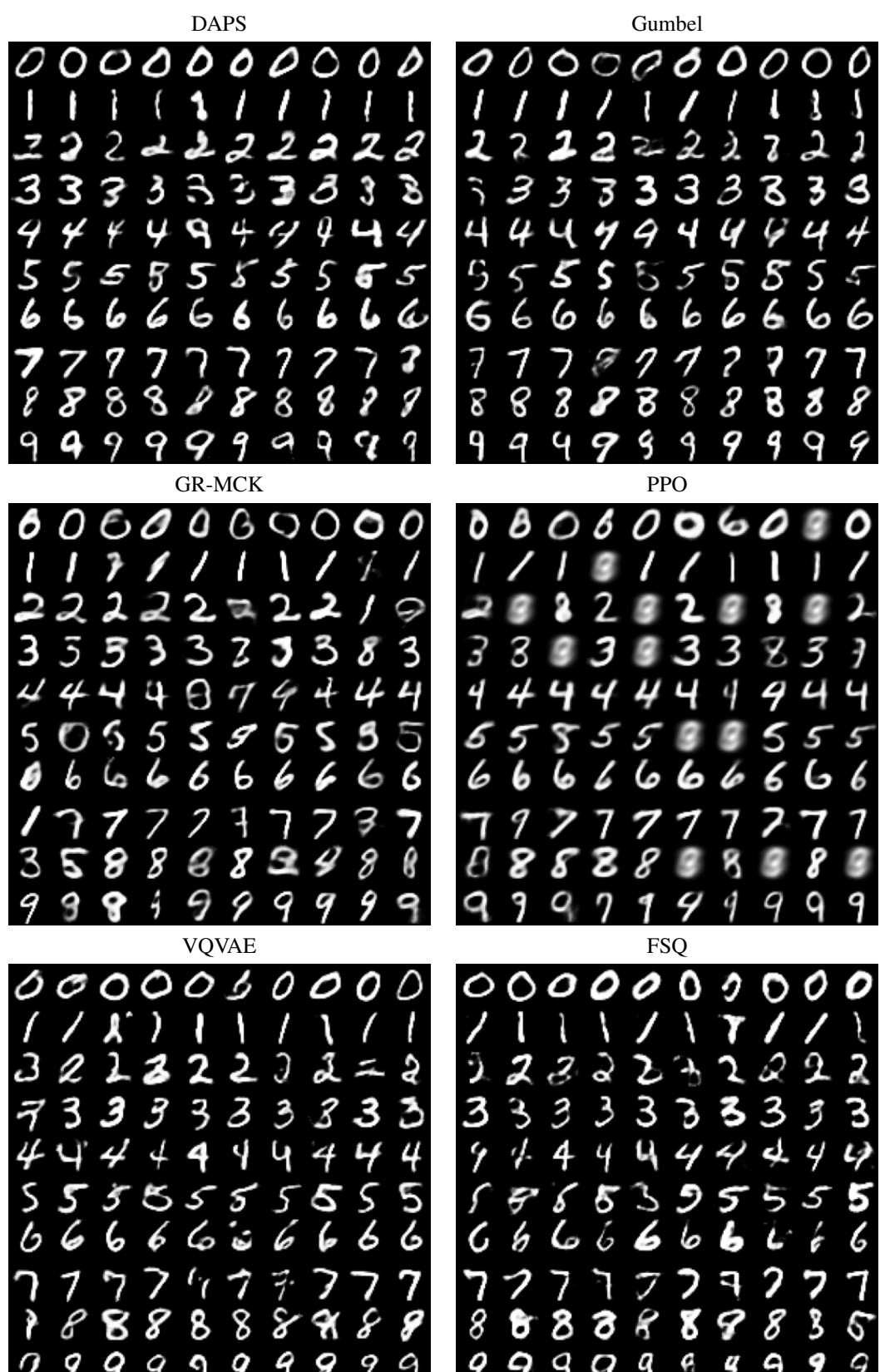

Figure 20: Label-conditioned samples from downstream model using discrete flow matching.

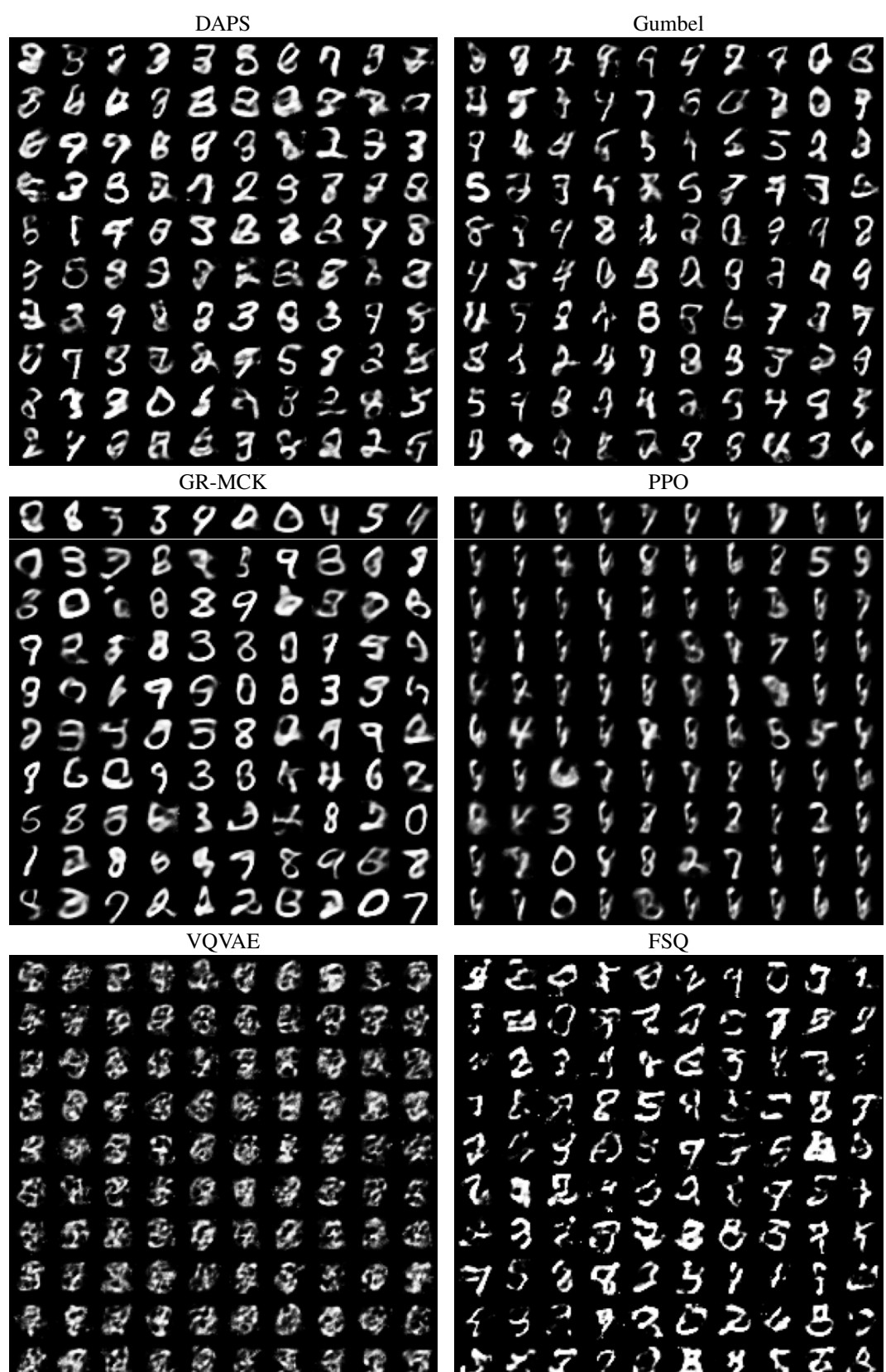

Figure 21: Samples obtained by decoding random, uniform prior latent codes.

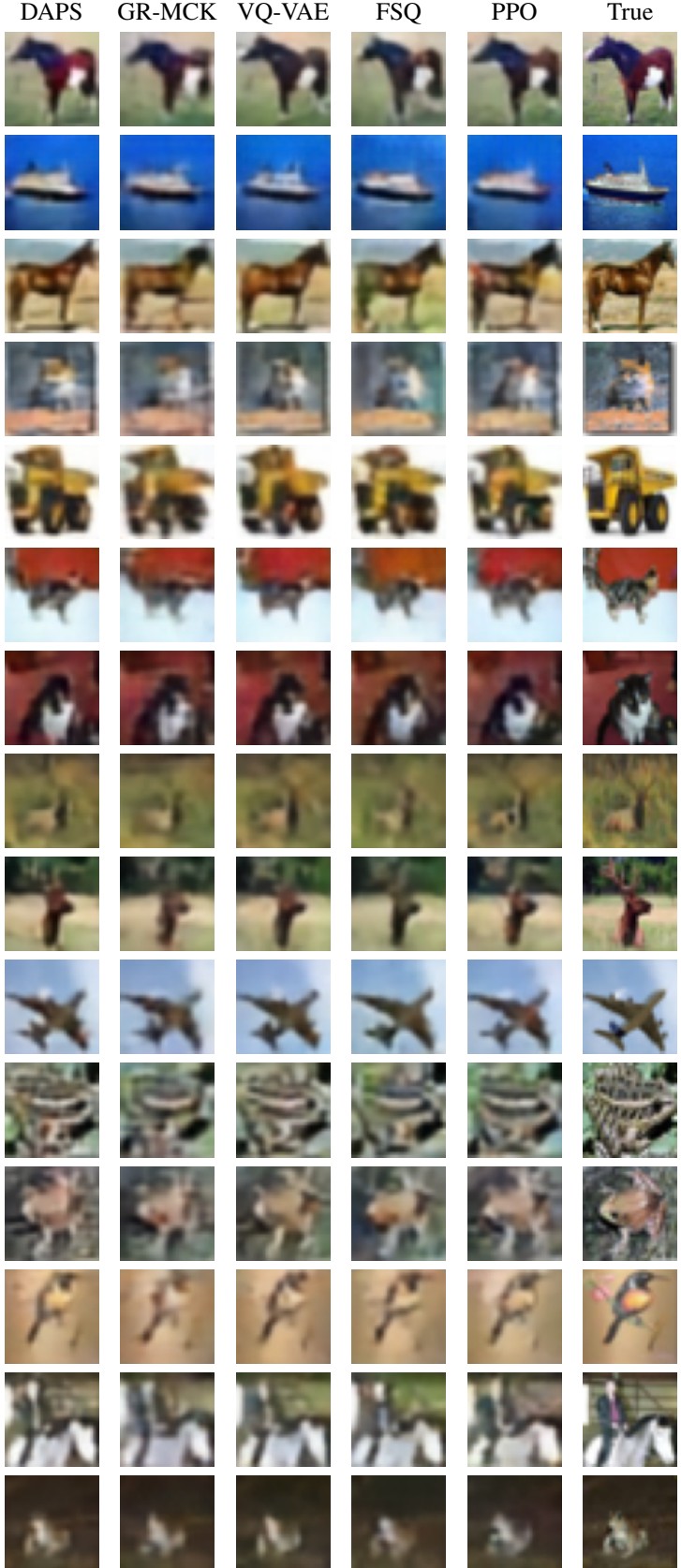

Figure 22: CIFAR validation reconstructions vs. ground truth.

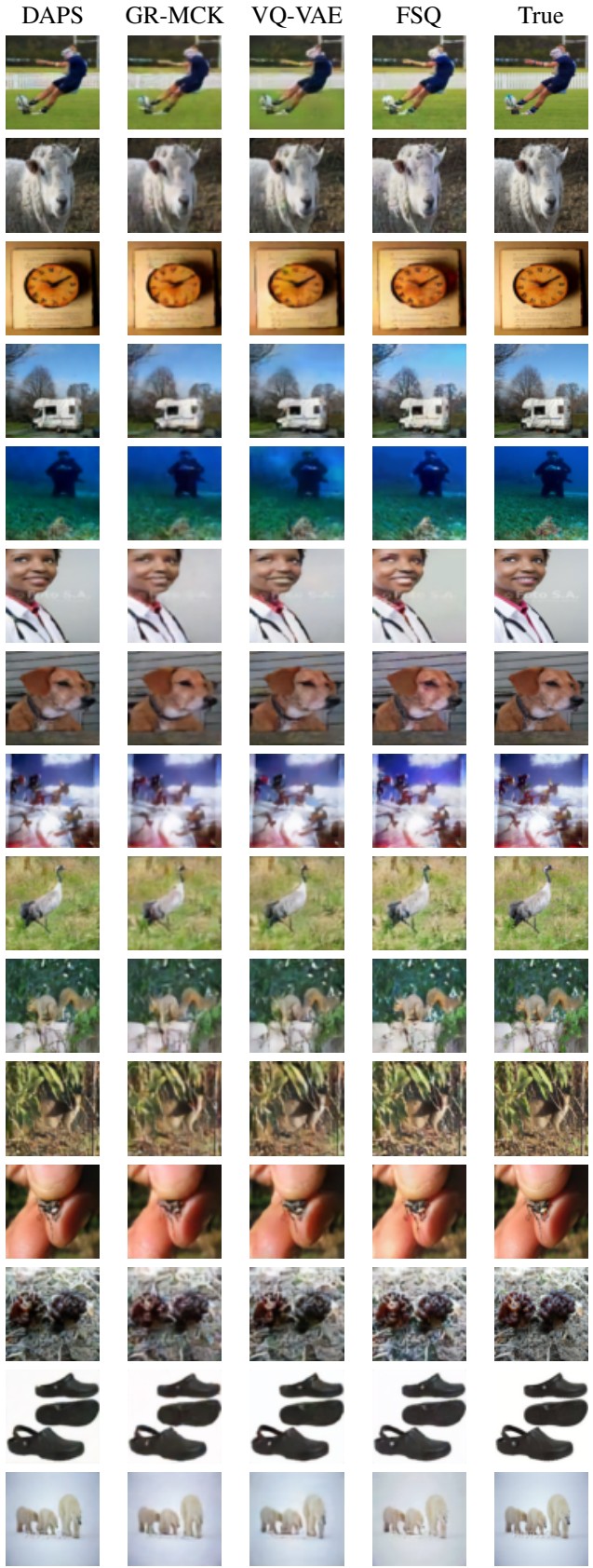

Figure 23: ImageNet 256 validation reconstructions vs. ground truth.

