# OpenReview forum: "Discrete Variational Autoencoding via Policy Search"
_ICLR.cc/2026/Conference — ICLR 2026 Poster_

### Official Review · Reviewer_i7u6 · 2025-10-20

**Soundness:** 2
**Presentation:** 3
**Contribution:** 3
**Rating:** 6
**Confidence:** 3

**Summary:**

This paper introduces Discrete Autoencoding via Policy Search (DAPS), a novel training framework for discrete VAEs that avoids reparameterization or straight-through gradient estimation. DAPS formulates encoder learning as a KL-regularized policy search problem, and the parametric encoder is updated via weighted maximum likelihood using importance sampling. A trust-region parameter is adapted automatically using the effective sample size (ESS) to ensure stable training. Experiments on MNIST, CIFAR-10, ImageNet-256, and LAFAN motion data show that DAPS outperforms Gumbel-Softmax, GR-MCK, FSQ, and VQ-VAE in reconstruction quality and bit efficiency.

**Strengths:**

1. DAPS is a principled training framework that reformulates discrete VAE encoder optimization as a KL-regularized policy search problem, enabling stable and scalable training without reparameterization. The methods leverage modern RL techniques that have been underexplored in discrete VAEs.
2. DAPS demonstrates state-of-the-art reconstruction quality on ImageNet-256, and outperforms GR-MCK and Gumbel-Softmax in both log-likelihood and FID across MNIST, CIFAR-10, and ImageNet. It scales efficiently to sequence lengths of 1,024.
3. The step size of DAPS adapts automatically via ESS, eliminating the need for manual tuning of the trust-region parameter.

**Weaknesses:**

1. The baselines involved in the experiments lack the most recent work in 2024/2025; therefore, it is not clear how DAPS compares to recent SOTA methods.
2. Section 3 states that DAPS avoids computationally expensive and high-variance backpropagation through time by using weighted maximum likelihood. However, the paper doesn't empirically quantify the computational savings compared to Gumbel-Softmax or GR-MCK.
3. The abstract claims a 20% improvement on FID  for ImageNet 256. It would be beneficial to have a hypothesis test to determine whether this improvement is significant.

**Questions:**

1. Section 5 mentions that DAPS generates coherent and high-quality motion trajectories for LAFAN, but Figure 3 only shows a single frame without quantitative motion quality metrics. Could you provide quantitative metrics comparing DAPS against baselines on motion-specific evaluation criteria, and clarify how 'coherence' was assessed (e.g., via human evaluation or physical plausibility checks)?

---

> ### Author Response · Authors · 2025-11-21
> **Initial Response for Reviewer 3**
>
> **Comment 3.1**
> Baselines lack recent work in 2024/2025...
>
> **Reply:**
> Thank you for the suggestion. We believe we have included the most relevant baselines within discrete autoencoding. It is not clear to us which additional recent methods would provide a meaningful comparison. We would appreciate it if the reviewer could indicate specific missing works.
>
> ---
>
> **Comment 3.2**
> The paper doesn't quantify computational savings...
>
> **Reply:**
> Thank you for raising this point. We believe that this is a valuable addition to our paper. We are adding an appendix section quantifying memory usage and speed relative to baseline methods, and will report these results soon.
>
> ---
>
> **Comment 3.3**
> Hypothesis test for ImageNet FID improvement...
>
> **Reply:**
> Thank you for bringing this up. Since we are rerunning baselines with matched parameter counts (see Reviewer 2 discussion), we will report statistical significance once the updated results are available.
>
> ---
>
> **Comment 3.4**
> Could you provide motion-specific quantitative metrics and clarify coherence?
>
> **Reply:**
> Thank you for raising this point. Our evaluation combines (i) the L2 reconstruction error over the full pose sequence and (ii) qualitative rollout analysis in the physics simulator. As shown in the supplementary videos, decoded motions remain temporally coherent in the sense that they track retargeted reference trajectories over time, match the intended behavior class (walking, running, dancing), and maintain stable whole-body coordination without jitter. The semi-transparent purple reference agent makes this correspondence visually explicit. Our goal here is not to benchmark a new control policy, but to demonstrate that the discrete latent representations learned by DAPS are expressive enough to drive complex, high-dimensional behaviors. The combination of pose trajectory error and simulator rollouts is a standard way to assess motion coherence in this setting.

---

> > ### Author Response · Authors · 2025-12-03
> >
> > We now updated our revision that includes all requested changes.
> >
> > __FID Score Hypothesis Test__: After incorporating the newly suggested non-autoregressive version of Gumbel (Gumbel-NA) into our experiments, we found that this method resulted in the closest performance to DAPS on ImageNet 256. To compare the performance of these two methods, we conducted a t-test between DAPS and Gumbel-NA, using the best validation FID scores for each seed (each seed representing an independent sample for the t-test). The results indicate a statistically significant difference, with a p-value less than alpha = 0.05. Additionally, we note that the best validation FID score across DAPS runs is 44.59, while the lowest obtained by baselines (Gumbel-NA) is 50.91, representing a 12.42\% improvement.
> >
> > __Computational comparison__: A comparison of runtime performance and resource usage for all methods has been added to the appendix. This includes the throughput (number of gradient steps per minute), GPU memory usage, and parameter counts for all methods.
> >
> > __Measuring the motion similarity__: We have added the PSNR to help quantify the difference between our reconstruction and ground truth. Since the squared distance is a standard metric in motion imitation settings ("DeepMimic", Peng et al.), and the PSNR is a deterministic transformation of the MSE, we believe this metric adds a useful perspective.

---

### Official Review · Reviewer_WViT · 2025-10-29

**Soundness:** 2
**Presentation:** 2
**Contribution:** 2
**Rating:** 4
**Confidence:** 5

**Summary:**

This paper introduces DAPS (Discrete Autoencoding via Policy Search), a novel method for training discrete variational autoencoders (VAEs) that leverages policy search techniques from reinforcement learning. The core idea is to frame the encoder update as a KL-regularized policy optimization problem, avoiding backpropagation through discrete sampling and enabling the use of autoregressive encoders. The method is evaluated on image (MNIST, CIFAR-10, ImageNet) and motion (LAFAN) datasets, demonstrating improved performance in reconstruction quality and downstream tasks like robot control.

**Strengths:**

The integration of policy search principles (like REPS and trust-region adaptation via ESS) into discrete VAE training is a novel and potentially impactful contribution.

The paper provides extensive experiments, showing that DAPS can scale to challenging datasets like ImageNet and outperform strong baselines such as Gumbel-Softmax, GR-MCK, FSQ, and VQ-VAE on key metrics (e.g., FID, log-likelihood).

The method demonstrates stable training.

**Weaknesses:**

The paper's logic is often indirect and difficult to follow. The connection between the core objective (discrete autoencoding) and the RL framework is not smoothly bridged, making the "big picture" and the intuitive justification for the approach unclear. In fact, both sides can be directly bridged via reverse KL minimization.

The notation is frequently confusing and inconsistent.
- A critical example is the description of the parametric encoder $q_{\theta}(z|x)$. It is initially presented as a standard categorical distribution, but later described as an autoregressive model decomposed into a product of conditional (categorical) distributions. This conflict is deeply confusing. Furthermore, Figure 1 visually suggests a non-autoregressive structure (e.g., $z_1$, $z_2$, $z_3$ are not shown as inputs for $z_4$), creating a direct inconsistency between the text, the formula, and the diagram.
- Eqs. (3)-(5) are presented with terminology and notations from reinforcement learning, despite Section 3 being mainly about discrete autoencoding. At least, the Advantage $A(z,x)$ should be more explicitly and clearly connected to the VAE objective.
- The recognition loss in Line 9 of Algorithm 1 is different from Eq. (6), particularly considering the gradient with respect to the parameter $\theta$.

The introduction of the constraint $D_{KL}[q(z|x) || q_{\theta}(z|x)] \le \epsilon_{\eta}$ in Eq. 4 actually changes the optimization from a reverse KL (in the standard ELBO) to a forward KL w.r.t. parameter $\theta$. Reverse KL is mode-seeking, while forward KL is mode-covering. The paper provides no discussion or analysis of this fundamental shift, its implications for the latent representation, or its potential to cause conflicting behaviors during training.

The inclusion of a "Goal-Conditioned Robot Control" experiment feels disconnected from the primary narrative of discrete autoencoding. The paper fails to establish a clear, compelling link between this experiment and the core contribution discrete autoencoding.

**Questions:**

In Table 1, were the encoder architectures for all baseline methods (Gumbel, GR-MCK, FSQ, VQ-VAE) identical to the one used for DAPS? If not, how can we be sure that the performance gains are due to the proposed training algorithm and not a more powerful or better-suited architecture? Please clarify this in the experimental setup.

Please see also the above Weaknesses.

---

> ### Author Response · Authors · 2025-11-21
> **Initial Response for Reviewer 2**
>
> **Comment 2.1**
> The paper's logic is often indirect and difficult to follow...
>
> **Reply:**
> Thank you for the constructive feedback. We revised Section 3 to provide a smoother bridge between discrete autoencoding and the policy-search formulation. We also appreciate your observation that reverse KL minimization can bridge these ideas. We believe the revised Section 3 makes the relationship between Equation 1 and Equation 3 clearer, as they are both reverse KL objectives.
>
> ---
>
> **Comment 2.2**
> Notation is confusing and inconsistent...
>
> **Reply:**
> Thank you for pointing this out. We now explicitly mention at the beginning of Section 2 that the latent variable can be written as a sequence. We agree that it was confusing not to explicitly state that the latent code $ \mathbf{z}=(\mathbf{z}_1,\dots,\mathbf{z}_T) $ is split into different blocks, which made it unclear why we introduced a joint distribution $q(\mathbf{z}) = q(\mathbf{z}_1) q(\mathbf{z}_2|\mathbf{z}_1)$ in the introduction and later consider $q(\mathbf{z}_1, ..., \mathbf{z}_N)$ (parameterized autoregressively). We also updated Figure 1 to clearly depict dependence on previously generated latent components during decoding. The revised figure, equations, and prose are now consistent.
>
> ---
>
> **Comment 2.3**
> Eqs. (3)-(5) are presented with terminology and notations from RL...
>
> **Reply:**
> We appreciate your feedback on this aspect and have revised Section 3 to avoid using reinforcement learning terminology prematurely. In particular, the advantage is now discussed in more detail, without assuming any prior knowledge about RL, and the reader is referred to the relevant works.
>
> ---
>
> **Comment 2.4**
> The recognition loss in Algorithm 1 appears inconsistent...
>
> **Reply:**
> The algorithm block has been updated to clarify the update of the parametric encoder. We apologize for the confusion regarding the sampling procedure and its relationship to the gradient of the encoder update. The stop-gradient operator is an implementation detail consistent with our derivations; please see the similar discussion with Reviewer 1.
>
> ---
>
> **Comment 2.5**
> The introduction of the constraint in Eq. 4...
>
> **Reply:**
> We would like to clarify that we are still optimizing a reverse KL objective given by the ELBO. The trust region KL constraint is used during the update of the non-parametric model to keep it close to the most recent parametric encoder, which is kept fixed while solving this optimization. Hence, the constraint corresponds to a reverse KL and is used to control the change during a single update (similar to a learning rate), not to change the optimization objective. Our approach uses an M-projection update of the parametric encoder to approximate $q^*$; however, this alternating procedure still optimizes the original ELBO objective, which can be derived from expectation-maximization ("Maximum a Posteriori Policy Optimisation", Abdolmaleki et al.) or approximate mirror descent ("Guided Policy Search as Approximate Mirror Descent", Montgomery and Levine).
>
> ---
>
> **Comment 2.6**
> The inclusion of the robotics experiment feels disconnected...
>
> **Reply:**
> The robotics experiment aims to show that DAPS applies to complex, high-dimensional tasks beyond the pixel-based domain. It also demonstrates that the learned latent representations can be effectively used by downstream models, e.g., controlling a robot based on a language command. We have added this motivation to the introduction.
>
> ---
>
> **Comment 2.7**
> In Table 1, were the encoder architectures for all baselines identical?
>
> **Reply:**
> Thank you for this suggestion. Despite our efforts to match transformer architectures as closely as possible, using equivalent decoder architectures introduces additional parameters for autoregressive methods (due to the transformer encoder). As a result, non-autoregressive methods are directly comparable in total parameters, while autoregressive methods ended up with more. We apologize for this oversight. We are rerunning experiments with parameter-matched VQ-VAE and FSQ decoders and will update the paper with the results. We have added the following clarification to the experimental setup: “Because VQ-based methods rely on a decoder-only transformer while ELBO-based methods use an encoder–decoder architecture, we increase the number of layers and hidden units in the VQ decoders to match their total parameter count as closely as possible to ours.”

---

> > ### Author Response · Authors · 2025-12-03
> >
> > We have now updated our revision to address all remaining concerns.
> >
> > __Clarifying parameter counts in all methods__: We have provided a listing of parameter counts for all methods in the appendix. Architecture diagrams and network variables are listed in the appendix for all methods and datasets. We have also updated the MNIST generative model, which initially was based on the architecture used by Kingma \& Welling ("Autoencoding Variational Bayes"), but led to an unnecessary large number of parameters compared to high-dimensional experiments. Upon making these adjustments (i.e., adding more layers to decoder-only architectures) to ensure that all methods have the same (or slightly more) parameters as DAPS, we re-ran the experiments for the affected methods. Our manuscript has been updated to reflect this change. We found that FSQ did not benefit from the increase in parameters, similar to the findings in the original paper. Generally speaking, we found that VQ-VAE saw a small improvement on ImageNet. Overall, our conclusions are consistent with those before; in light of these adjustments, DAPS still outperforms the baseline methods.
> >
> > __Notation__: We updated the manuscript to address the notational concerns, particularly regarding the factorization of the categorical variable into conditionals (implemented as an autoregressive transformer in practice), ensuring consistency across all figures and equations.
> >
> > __Equation 4 Constraint__: We have clarified that the trust region constraint is a reverse KL divergence with respect to the distribution being optimized, rather than a forward KL divergence. This, however, does not alter our original (reverse KL) objective of optimizing the ELBO; rather, it influences the step size of our update, similar to a learning rate.

---

### Official Review · Reviewer_tBAf · 2025-10-30

**Soundness:** 2
**Presentation:** 2
**Contribution:** 3
**Rating:** 4
**Confidence:** 3

**Summary:**

The main challenge in training discrete VAEs lies in backpropagating through the discrete sampling process. In this work, the authors propose Discrete Autoencoding Policy Search (DAPS), a novel training framework for discrete VAEs inspired by policy search methods from reinforcement learning, particularly Relative Entropy Policy Search (REPS). The method exploits the gradient of a non-parametric encoder (which can be optimized in closed form but not directly sampled) to update the parametric encoder without relying on reparameterization, while also incorporating automatic step-size adaptation. This approach eliminates the need for differentiable soft relaxations or zero-order gradient estimators, which often lead to biased or high-variance gradients and unstable training.

**Strengths:**

Given the increasing interest in learning discrete representations in the literature, I believe exploring new methods that enable the effective training of discrete latent variable models is relevant. While connections with reinforcement learning have previously been explored to address the challenge of backpropagating through discrete sampling (most notably through methods like REINFORCE) to the best of my knowledge, this is the first work to apply particle-based policy search methods to train discrete VAEs. This approach effectively circumvents the reparameterization problem and enables more stable training procedures. The authors also show promising results on complex, high-dimensional datasets, additionally highlighting the potential of this method for broader applicability in downstream robotics tasks.

**Weaknesses:**

-	**Method description.** Overall, I believe the description of the method could be improved. While some aspects may be intuitive for readers with a reinforcement learning background, they may appear opaque to others with expertise in deep generative models but not reinforcement learning (such as myself). For instance, why is the advantage function used instead of the reward function? In Equation (5), the loss for updating the encoder involves sampling from the parametric posterior, why does this not introduce the same backpropagation issue? I noticed that the training algorithm includes a stop-gradient operator, but I believe this should be explicitly discussed in the main text, since a central aspect of the method is preventing backpropagation through the discrete sampling process.

-	**Writing.** Additionally, I think the overall writing could be improved. In the introduction, several acronyms (e.g., ELBO) are not defined until later sections, and some variables referenced in the algorithm are never described in the text. Likewise, the metrics reported in the tables are not defined (e.g., log p, L2).

-	**Method.** While I find the framework interesting, it's unclear to me the ratio between performance gain and computational efficiency, as it is something that the authors do not address. They mention efficiency in terms of memory relative to Gumbel-Softmax-based methods, but not with respect to other baselines, nor in terms of training time. Also, as I stated before, I think key aspects of the method, such as the sampling procedure, remain unclear to me after reading the paper. I would appreciate if the authors can clarify it during the rebuttal.

-	**Evaluation.** I find the evaluation of the method limited. While the authors provide some reconstruction examples, showing that qualitatively their method performs comparably to state-of-the-art approaches, they do not include quantitative metrics to assess reconstruction quality, such as PSNR, comparing original and reconstructed samples. For generation, although FID scores are reported for some methods, the sample size used is not specified, which is important for comparison with state-of-the-art results. Additionally, it is unclear why these metrics could not be obtained for certain model configurations. Regarding the latent representations, the authors do not analyze their use. This is a notable limitation of VQ-VAEs that FSQ aims to address. It would be interesting to investigate how this alternative training framework, which allows for more stable training, impacts posterior inference. Furthermore, it would be valuable to study how the dimensionality of the latent space influences overall performance (could the improved training allow for more compact latent spaces?), especially since the latent representations appear to be high-dimensional. Also, I think the use of the ELBO as evaluation metric can be misleading, for example, a model with posterior collapse ELBO may still be high because reconstruction is good, but the latent code carries no information.

**Questions:**

-	Would this method still be beneficial when considering simpler models without an autoregressive structure, and a single vector as latent representation? One of the motivations of this approach is to improve memory efficiency with respect to approaches such as Gumbel-Softmax, does the same argument apply in simpler scenarios?
-	In this context, is generation performed by sampling from the uniform prior assumed during training? Similar to VQ-VAE, does this method require training an autoregressive prior in a second stage? How is the generation process?
-	In the appendix, the authors present label-conditioned samples. How is label-conditioned generation implemented? In the downstream robotics example, the high-level policy is also conditioned on language information, how is this text-conditioning realized in the DAPS generative process?
-	According to the authors, what are the main limitations of the method, and what are the primary directions for future research?


I would be happy to increase my score if the authors adequately address the main weaknesses and questions.

---

> ### Author Response · Authors · 2025-11-21
> **Initial Response for Reviewer 1**
>
> **Comment 1.1**
> Overall, I believe the description of the method could be improved. While some aspects may be intuitive for readers with a reinforcement learning background, they may appear opaque to others with expertise in deep generative models but not reinforcement learning (such as myself). For instance, why is the advantage function used instead of the reward function?
>
> **Reply:**
> Thank you for this thoughtful suggestion. We have revised Section 3 to take this feedback into account. In particular, we removed any undefined reinforcement learning terminology, as we understand this may have caused more confusion than help. Regarding the use of the advantage function, we have dedicated a small section to explain its role.
>
> ---
>
> **Comment 1.2**
> In Equation (5), the loss for updating the encoder involves sampling from the parametric posterior. Why does this not introduce the same backpropagation issue? I noticed that the training algorithm includes a stop-gradient operator, but I believe this should be explicitly discussed in the main text.z
>
> **Reply:**
> Thank you for pointing out this inaccuracy in Equation 5. This equation describes the maximum likelihood projection of the non-parametric distribution $q^*$ onto our parametric encoder $q_\theta$. As such, the parametric model originally only appears as a log-likelihood, but it is not used for sampling. However, as we cannot sample from the non-parametric model, we use a frozen copy of the parametric model to draw samples and use importance weighting to correct for this mismatch. In the original submission, it seemed like $\boldsymbol{\theta}$ should affect the sampling process, the importance weights, and the log-likelihood term. We fixed this issue in our revision by clarifying that only the log-likelihood term depends on $\boldsymbol{\theta}$. The use of stop-gradient in the algorithm box is fully in line with our derivations and is a convenience in the implementation to avoid explicitly creating a copy of the parametric model for creating the dataset and importance weights for the maximum likelihood update.
>
> ---
>
> **Comment 1.3**
> The writing could be improved. Some acronyms (e.g., ELBO) are not defined until later sections, variables in the algorithm are not described, and metrics such as log p and L2 are not defined.
>
> **Reply:**
> Thank you for pointing this out. We updated the manuscript to define all acronyms at first occurrence (e.g., method names and $\beta$) and added explicit definitions for all reported metrics (L2, log $p(x|z)$, and $\beta$-ELBO) in Section 5. We also revised Algorithm 1 for improved clarity. Additionally, as part of our effort to improve clarity, we ensure that the RL variables are defined in a more consistent way.
>
> ---
>
> **Comment 1.4**
> While I find the framework interesting, it's unclear to me the ratio between performance gain and computational efficiency, as it is something that the authors do not address. They mention efficiency in terms of memory relative to Gumbel-Softmax-based methods, but not with respect to other baselines, nor in terms of training time.
>
> **Reply:**
> Thank you for raising this point. We agree that comparing memory usage and training speed across baselines is important. We have benchmarked DAPS, Gumbel, GR-MCK, FSQ, and VQ-VAE in terms of GPU memory footprint and throughput; these runs are still being consolidated and double-checked. We will include the full quantitative comparison in the updated appendix.
>
> ---
> **Comment 1.5**
> Key aspects of the method, such as the sampling procedure, remain unclear to me after reading the paper. I would appreciate if the authors can clarify it during the rebuttal.
>
> **Reply:**
> We have revised Algorithm 1 and Equation 6 to make the sampling and importance-weighting procedure explicit, and we corrected Equation 6 to match the estimator used in practice. We believe these changes resolve the ambiguity, and we thank the reviewer for highlighting this gap.

---

> > ### Author Response · Authors · 2025-11-21
> > **Continued...**
> >
> > **Comment 1.6**
> > Limited evaluation (no PSNR comparisons); unclear sample size for FID; unclear why some metrics are missing.
> >
> > **Reply:**
> > We appreciate the reviewer’s very constructive suggestions. This has motivated a new series of ablations that are already underway. Please note that, in light of the helpful suggestion from Reviewer 2, we have decided to rerun VQ-based methods with more parameters in an effort to make our architectures more comparable. For this reason, the following evaluations will be conducted using the new models.
> >
> > (i) **Reconstruction metrics.** We are computing PSNR and SSIM across the different vision tasks. These results will be posted once verified. Our motivation for using the squared reconstruction loss (L2) was to have a general metric for all datasets (including LAFAN); however, we are happy to report PSNR instead, as it carries similar information (being a transformation of the MSE).
> >
> > (ii) **FID sample size.** We now explicitly state in the paper that the FID score for ImageNet corresponds to the full validation dataset (50k samples) and the full validation set for CIFAR-10 (10k samples). This has been added to the first paragraph of the results section.
> >
> > (iii) **Missing metrics for certain baselines.** The reasons for omitting Gumbel and PPO are briefly stated in Section 4 of the experiments. On ImageNet-256, Gumbel-Softmax was unstable and consistently diverged despite temperature tuning and learning-rate scheduling; PPO likewise failed to reach competitive likelihoods at this scale. For space reasons, some secondary metrics (e.g., L2 on CIFAR/ImageNet) were moved to the figures in the appendix; in the revision we will report a more uniform metric set in Table 1 and keep additional curves in the appendix.
> >
> > ---
> > **Comment 1.7**
> > Regarding the latent representations, the authors do not analyze their use...
> >
> > **Reply:**
> > We agree that this is an important aspect, given the known limitations of VQ-VAE. We are adding a new analysis of the codebook usage to help characterize the latent utilization of all methods. While the entropy of VQ-based methods cannot be computed in the traditional sense (and therefore cannot be directly compared with our method), a sample-based analysis of latent diversity on the validation set will help characterize utilization across methods.
> >
> > ---
> >
> > **Comment 1.8**
> > It would be valuable to study how the dimensionality of the latent space influences overall performance...
> >
> > **Reply:**
> > Motivated by this comment, we are conducting new scaling experiments: a “tiny”, “small”, and “medium” bottleneck setting for MNIST and CIFAR-10, each with a full hyperparameter sweep and 5 seeds. These experiments allow us to quantify how the compression level affects performance. We will upload the full results soon.
> >
> > ---
> >
> > **Comment 1.9**
> > I think the use of the ELBO as an evaluation metric can be misleading...
> >
> > **Reply:**
> > Thank you for this suggestion. We believe that the latent code utilization experiment will help assess whether the latent codes carry meaningful information. We will include a discussion of collapse/usage once those results are finalized. However, although ELBO cannot capture all relevant aspects, we argue that it is still an interesting metric, as it is the objective that VAEs aim to optimize.
> >
> > ---
> >
> > **Comment 1.10**
> > Would the method still be beneficial when considering simpler models without an autoregressive structure?
> >
> > **Reply:**
> > Thank you for the insightful question. We are currently evaluating DAPS using a non-autoregressive (NA) decoder-only model (similar to VQ-based methods), which outputs a distribution over the latent vector. While these experiments are running, our initial observations suggest that DAPS-NA provides a computational speedup at the expense of performance. We will include the full results as soon as they are finalized. Indeed, the computational advantage with respect to Gumbel-Softmax diminishes in this situation; we will quantify this difference in the memory/speed comparison in our appendix. A key motivation for DAPS is its ability to model general joint distributions over latent variables through an autoregressive structure (something that VQ-VAE/FSQ encoders cannot express). For example, the encoder output could respect constraints such as “either $z_1=z_2=1$ or $z_1=z_2=0$, but not mixed configurations”. We hypothesized that this additional expressiveness could help organize the latent space during early optimization, as it allows the encoder to remain less committal. However, while the autoregressive variant indeed achieves stronger performance in our experiments, we have not yet found clear evidence supporting this specific mechanism.

---

> > > ### Author Response · Authors · 2025-11-21
> > > **Continued...**
> > >
> > > **Comment 1.11**
> > > In this context, is generation performed by sampling from the uniform prior assumed during training?
> > >
> > > **Reply:**
> > > To evaluate the learned latent space, we sample from the latent prior used during training (a discrete uniform distribution). These randomly sampled latents are then decoded and visualized, as shown in Figure 14. Therefore, we do not use an autoregressive prior. We would like to note that the use of a powerful autoregressive prior employed in VQ-VAE often requires substantial computational resources; on the other hand, decoding randomly sampled latents does not yield sharp generative samples beyond low-dimensional settings due to the combinatorial complexity of the latent space.
> > >
> > > ---
> > >
> > > **Comment 1.12**
> > > How is label-conditioned generation implemented?
> > >
> > > **Reply:**
> > > In the second stage, we apply a discrete flow matching (DFM) procedure, which closely parallels the mechanism employed in MaskGIT. We begin by sampling a latent vector from a uniform prior and train a DFM process that maps this sample to one from the distribution of $q(z|x)$, given a label. This is done in a supervised learning setting: we train on the fixed dataset generated from our recognition model (regression targets) alongside the corresponding class labels (regression inputs).
> > >
> > > ---
> > >
> > > **Comment 1.13**
> > > How is text-conditioning implemented in the robotics example?
> > >
> > > **Reply:**
> > > For each motion in the LAFAN dataset, we have a brief description (e.g., “walking subject 1” or “dancing subject 2”) that is fed into a pretrained BERT tokenizer. Once tokenized, this context is provided, along with an encoding of the proprioceptive history and desired future center of mass trajectory, to a second-stage model. The second-stage model is trained to output the same latent code as the one encoded by DAPS for the particular motion segment associated with this context (see Appendix Section I). In this setting, we make effective use of (i) the pretrained DAPS latent space and (ii) a physically consistent decoder to synthesize motions conditioned on a desired behavior. This task highlights that the learned latents can drive multimodal whole-body behaviors in a physically consistent simulator.
> > >
> > > ---
> > >
> > > **Comment 1.14**
> > > What are the main limitations and primary directions for future research?
> > >
> > > **Reply:**
> > > The main limitations are currently the gap in computational performance between autoregressive DAPS and non-autoregressive methods such as VQ-VAE. While there is a clear trade-off between performance and computational complexity, improving this aspect would further strengthen the method. Possible future directions include applying DAPS to language-conditioned tasks, where latent encodings can provide useful contexts for downstream applications.

---

> > > > ### Comment · Reviewer_tBAf · 2025-11-27
> > > >
> > > > I would like to thank the authors for their response and for their efforts in conducting additional experiments and improving the clarity of the manuscript. Since some of my concerns will be addressed in the revised version (including clarifications, corrections and new experimental results), I will wait for the updated PDF before providing my final assessment. I would greatly appreciate it if the authors could clearly highlight the changes in the revised version.

---

> > > > > ### Author Response · Authors · 2025-12-03
> > > > >
> > > > > Thank you for your response. We have created a PDF with the changes and added it to the supplementary. We have now also added all additional results.
> > > > >
> > > > > __Investigation of simpler, autoregressive models__: We have conducted an extensive study to evaluate the use of simpler, non-autoregressive (NA) variations of ELBO-based methods. For each of the non-autoregressive methods, we performed a hyperparameter search over the model-specific parameters and investigated their performance in both low-entropy (beta = 0.01) and standard (beta = 1.0) settings. These results have been included in our main findings for all datasets, using the same number of seeds as the original methods. We find that DAPS outperforms these non-autoregressive models, and that the non-autoregressive version of Gumbel Softmax generally works better than its autoregressive counterpart. We believe this variation can help alleviate the difficulties of gradient estimation in autoregressive sampling. Despite this, neither variation of the Gumbel Softmax consistently outperformed VQ-VAE or FSQ; on the other hand, both DAPS and DAPS-NA consistently exhibit strong performance across all datasets.
> > > > >
> > > > > __Investigation of latent usage__: We have added experiments to analyze the use of the latents, including a small section in the paper and a more extensive section in the appendix. In particular, we report the empirical distribution of the marginal codebook/vocabulary usage across three different seeds for MNIST and CIFAR. This helps depict how frequently individual codes are used when encoding the full validation dataset. We find that DAPS exhibits strong use of the latent vocabulary, having highly uniform code usage. We observe superior codebook utilization compared to FSQ and VQVAE, despite FSQ's attempts to address this issue. This finding further highlights that the latent space learned by DAPS has desirable entropy properties originally optimized for.
> > > > >
> > > > > __Improved consistency in our main table__: We have updated the metrics in Table 1 to include PSNR. We appreciate this suggestion, as the PSNR provides a standard medium for comparing reconstruction quality across different datasets and related works. To provide a more accurate estimate of the PSNR value (which utilizes normalized features) for LAFAN, we exclude the 6-dimensional root velocity feature due to its adverse impact on normalization (and low importance in imitating the motions). This ultimately reduced the dimension of LAFAN to 32x121 from 32x127, and we reran all LAFAN experiments to account for this. We also resolved the issue of missing metrics for some method/dataset combinations. Our table is now split more appropriately. We use two compact tables to emphasize the different objectives for each study. An explanation of this setup is provided in the results section to help guide the reader.
> > > > >
> > > > > __Runtime/Resource comparison__: A comparison of runtime performance and resource usage for all methods has been added to the appendix. This includes the throughput (number of gradient steps per minute), GPU memory usage, and parameter counts for all method variations (non-autoregressive  + autoregressive). We find that the non-autoregressive version of DAPS yields better resource utilization and speed than autoregressive DAPS, albeit at the expense of reduced performance. We find that DAPS is the most efficient among the class of autoregressive models in terms of memory and speed. While DAPS is slower than the non-autoregressive methods, we argue that the increased computational costs are justified by the improved performance.
> > > > >
> > > > > __Scaling Comparison__: We conducted an experiment to assess the scaling properties of DAPS with respect to VQ and Gumbel variations on both MNIST and CIFAR datasets. For each setting (tiny, small, medium), we performed a hyperparameter search and reported results from 5 different seeds. The results indicate that the performance of DAPS scales with the latent capacity and tends to increase its performance gap as the bottleneck dimension increases.
> > > > >
> > > > > Prior to conducting the above investigations—and in response to reviewer 2's concerns—we have ensured that all methods use a number of parameters that is greater than or equal to our proposed method, DAPS. Thus, all reported metrics are consistent with these updates.

---

### Author Response · Authors · 2025-11-21
**Initial Response**

We sincerely appreciate the reviewers’ time and thoughtful feedback. We are very pleased with the insights provided and value the different perspectives offered. We agree with most of the concerns raised regarding the writing. We have improved the clarity of the paper, particularly in relation to the RL component and its connection to variational autoencoding. We genuinely believe that the quality of the paper has already improved thanks to the reviewers’ insights. We also agree that the requested additional ablations will strengthen the contribution and further enhance the clarity of the paper. These experiments are already running, and we will submit a revised manuscript including the improved writing and the additional results shortly. We greatly appreciate the reviewers’ patience. In the meantime, we address most of the questions below and would be happy to engage in further discussion. We would like to note that, in light of the new experiments underway, our results may change, and we will ensure that their implications are clearly presented.

---

### Author Response · Authors · 2025-12-03
**Response To Area Chair**

Dear newly assigned AC,

In the following, we aim to provide a brief summary of our contributions, the initial reviews, and our subsequent revisions.

## Contributions
We introduce a novel method for training VAEs with discrete bottlenecks, focusing in particular on autoregressive models, such as transformers, which can model the full joint distribution over the latent code. Our method avoids the use of approximate techniques, such as Gumbel-SoftMax or straight-through gradient estimation. Instead, we train the encoder using a technique from policy search methods, by using a two-stage approach, where we first update the weights of a particle-representation of the encoder, and then perform an importance weighted maximum likelihood update to realize the update step on the parametric model. Furthermore, we propose to perform a trust-region update on the particle-based encoder based on the effective sample size computed from the importance weights. Our experiments show that these techniques enable us to train autoregressive, discrete VAEs even on challenging data sets, such as ImageNet-256 or motion trajectories of humanoid robots, and achieve state-of-the-art performance even compared to popular methods such as VQ-VAE or FSQ that can not optimize the traditional ELBO objective, due to their inability to evaluate the encoder's entropy.

## Initial Reviews

### **Reviewer 1 (tBAf)**
Reviewer tBAf acknowledged the novelty of applying advanced policy search techniques to variational auto-encoding and recognized our success in effectively circumventing reparameterization issues of discrete VAEs while showing promising results on complex datasets.

The reviewer initially rated our work with a 4 due to concerns regarding the writing, method description, and the soundness of our approach, specifically noting the presence of a stop-gradient operation and a loss that involved sampling from the parametric encoder.

We have significantly revised the presentation of our method (please refer to the diff in the supplementary material) to completely address these concerns. Regarding soundness, we now explicitly clarify that our approach treats a frozen copy of the parametric model as a proposal distribution for importance-weighted maximum likelihood estimation. Consequently, we do not need to backpropagate through the sampling process, and the original equation was inaccurate. The inclusion of the stop-gradient operation in the algorithm box is an implementation convenience that is fully consistent with our theoretical derivations. This clarification should entirely dispel the concern regarding the soundness of our method.

The reviewer also requested an analysis of the ratio between performance gain and computational efficiency, the inclusion of PSNR as an additional metric, and an investigation into scaling to different bottleneck sizes. We have incorporated all requested experiments into the revision.

We now provide a detailed comparison of computational cost. When restricted to methods that learn the full latent joint distribution (i.e., autoregressive encoders), our method, DAPS, trains faster and occupies less memory. While all autoregressive models, including DAPS, have a higher computational cost compared to non-autoregressive models, our experiments consistently show that DAPS achieves superior performance while scaling sufficiently well for high-dimensional data sets. We have also added the PSNR metric and included new experiments investigating the effect of different bottleneck sizes.

By significantly improving the presentation of our work, dispelling all concerns regarding the soundness of the method (specifically the stop-gradient operation), and thoroughly addressing the requests for additional experiments, we believe we have fully satisfied the reviewer. **As the reviewer explicitly mentioned they would be happy to increase their score in this case, we are confident that they would advocate towards accepting our submission.**

---

> ### Author Response · Authors · 2025-12-03
> **Response To Area Chair (Continued)**
>
> ### **Reviewer 2 (WViT)**
> Reviewer WViT similarly acknowledged our method's novelty, potential impact, and performance against strong baseline methods. However, this reviewer left a 4 due to initially unresolved clarifications.
>
> In particular, this reviewer noted that the connection between RL and the VAE was not clear to readers. We have revised the manuscript to make this connection more explicit. In particular, equation 1 (the ELBO objective) can be rewritten as the episodic maximum entropy RL objective (equation 3), using our definition of the return. This approach of bridging the two topics avoids the assumption that the reader is familiar with RL terminology and also makes the reverse KL connection clearer (as both equations 1 and 3 are reverse KL objectives). The reviewer also criticized that we initially introduced the encoder as a standard categorical distribution, but later consider an auto-regressively modeled joint distribution. We now clarify in our revision that we divide the latent variable $\mathbf{z}$ into $B$ blocks $z=(z\_1, \dots, z\_B)$. Furthermore, similar to reviewer tBAf, reviewer WViT was concerned about the mismatch between our algorithmbox and the equations, which we addressed in our revision.
>
> Second, Reviewer 2 thought that the trust region constraint, which we use for updating the non-parametric encoder, would modify the ELBO objective, turning it into a maximum likelihood problem (forward KL minimization instead of reverse KL minimization). We clarified in our response that this trust region constraint merely controls the stepsize of the non-parametric update. It is also evident from our evaluations that our method is effective in optimizing the original ELBO objective. We also clarified that, although our method does indeed use a maximum likelihood approach to update the parametric model, this optimization serves as a projection to realize the non-parametric update on the parametric model.
>
> Hence, we addressed all concerns of reviewer 2, including the critical concerns that our method might no longer optimize the ELBO objective, and that our algorithmbox is not aligned with derivations. **Consequently, we are also confident that reviewer WViT would increase their score.**
>
> ### **Reviewer 3 (i7u6)**
>
> Reviewer 3 was initially already in favor of accepting our work, and mainly expressed concerns regarding (i) the lack of computational comparisons, (ii) a missing hypothesis test for the FID score, and (iii) claims of coherent motions.
>
> We have addressed all three points:
>
> i) A computational comparison has been added to our appendix (including a benchmark of the speed, memory usage, and parameter counts).
>
> ii) The newly added Gumbel-NA version obtained the closest performance to DAPS on ImageNet 256. Consequently, we applied the hypothesis test to assess whether the FID scores obtained by DAPS are statistically significant compared to those of Gumbel-NA. We found that DAPS is indeed statistically significant in comparison, having a p-value of less than 0.05. The resulting improvement is 12.4 percent (among the best scores of all seeds); however, this has been removed from the abstract.
>
> iii) We clarify that the supplementary videos demonstrate that the reference trajectories match the intended behavior class (walking, running, dancing) and maintain stable whole-body coordination without jitter, as visualized by side-by-side video comparisons. Furthermore, we argue that the squared error is a common metric for assessing the similarity between motions in robotics, similar to the exponentiated squared difference used by "DeepMimic" (Peng et al.). **We are confident that Reviewer 3 is now even more positive about submission.**

---

### Meta-Review · Area_Chair_joGu · 2026-01-06

**Summary:**

This manuscript introduces a  method for training discrete variational autoencoders (VAEs), termed DAPS, Discrete Autoencoding via Policy Search. DAPS that leverages policy search techniques from reinforcement learning, formulating the encoder update as a KL-regularized policy optimization problem and thereby avoiding backpropagation through discrete sampling. DAPS is evaluated on MNIST, CIFAR-10, ImageNet and LAFAN in terms of reconstruction quality and downstream task performance. Initial scores were borderline.
Reviewers agree that the problem of learning discrete representations is timely and therefore, it makes sense to explore alternatives to full backpropagation. The concerns are mostly focus on the paper presentation and some experiments and ablations. The authors have updated their manuscript according to the comments and included more results. After the discussion and the revision, I think this is a good paper with interesting contribution.

**Reviewer Concerns:**

Reviewer i7u6: The baselines involved in the experiments lack the most recent work in 2024/2025 (comment is slightly vague, hard to address)
Reviewer i7u6: the paper doesn't empirically quantify the computational savings compared to Gumbel-Softmax or GR-MCK (addressed)
Reviewer i7u6: statistical significance (addressed)
Reviewer tBAf: writing (partly addressed)
Reviewer tBAf: evaluation of the ratio between performance gain and computational efficiency (addressed)
Reviewer tBAf: evaluation of the method limited (at least partly addressed)
Reviewer WViT: Paper organization and writing (at least partly addressed)

I believe the most important concerns have been addressed and the authors have tried to improve the manuscript according to the reviewers guidance. I think it would have been really helpful to see the full discussion process for this paper, in particular to help the authors in improving their manuscript according to the reviewers suggestions.

**Reviewer Scores:**

With the provided rebuttal and revision, I would expect the review scores to consistently improve to 6.

---

### Decision · Program_Chairs · 2026-01-26

Accept (Poster)